EMBO
Molecular Medicine

# Flower lose, a cell fitness marker, predicts COVID-19 prognosis

Michail Yekelchyk[1,†] (ID), Esha Madan[2,†], Jochen Wilhelm[3,4,‡], Kirsty R Short[5,‡], António M Palma[2,‡], Linbu Liao[6,‡] (ID), Denise Camacho[2] (ID), Everlyne Nkadori[7], Michael T Winters[8], Emily S Rice[8] (ID), Inês Rolim[2], Raquel Cruz-Duarte[9], Christopher J Pelham[10], Masaki Nagane[11] (ID), Kartik Gupta[12] (ID), Sahil Chaudhary[12], Thomas Braun[1,13] (ID), Raghavendra Pillappa[14] (ID), Mark S Parker[15], Thomas Menter[16], Matthias Matter[16], Jasmin Dionne Haslbauer[16] (ID), Markus Tolnay[16], Kornelia D Galior[7] (ID), Kristina A Matkwoskyj[7], Stephanie M McGregor[7], Laura K Muller[7], Emad A Rakha[17], Antonio Lopez-Beltran[2,18] (ID), Ronny Drapkin[19,20,21] (ID), Maximilian Ackermann[22,23] (ID), Paul B Fisher[24,25,26], Steven R Grossman[27,28], Andrew K Godwin[29,30], Arutha Kulasinghe[31] (ID), Ivan Martinez[8], Clay B Marsh[8], Benjamin Tang[32], Max S Wicha[33,34], Kyoung Jae Won[6,35] (ID), Alexandar Tzankov[16,*] (ID), Eduardo Moreno[2,**] (ID) & Rajan Gogna[2,6,35,***] (ID)

## Abstract

Risk stratification of COVID-19 patients is essential for pandemic management. Changes in the cell fitness marker, *hFwe-Lose*, can precede the host immune response to infection, potentially making such a biomarker an earlier triage tool. Here, we evaluate whether *hFwe-Lose* gene expression can outperform conventional methods in predicting outcomes (e.g., death and hospitalization) in COVID-19 patients. We performed a post-mortem examination of infected lung tissue in deceased COVID-19 patients to determine *hFwe-Lose*'s biological role in acute lung injury. We then performed an observational study ($n = 283$) to evaluate whether *hFwe-Lose* expression (in nasopharyngeal samples) could accurately predict hospitalization or death in COVID-19 patients. In COVID-19 patients with acute lung injury, *hFwe-Lose* is highly expressed in the lower respiratory tract and is co-localized to areas of cell death. In patients presenting in the early phase of COVID-19 illness, *hFwe-Lose* expression accurately predicts subsequent hospitalization or death with positive predictive values of 87.8–100% and a negative predictive value of 64.1–93.2%. *hFwe-Lose* outperforms conventional inflammatory biomarkers and patient age and comorbidities, with an area under the receiver operating characteristic curve (AUROC) 0.93–0.97 in predicting hospitalization/death. Specifically, this is significantly higher than the prognostic value of combining biomarkers (serum ferritin, D-dimer, C-reactive protein, and neutrophil–lymphocyte ratio), patient age and comorbidities (AUROC of 0.67–0.92). The cell fitness marker, *hFwe-Lose*, accurately predicts outcomes in COVID-19 patients. This finding demonstrates how tissue fitness pathways dictate the response to infection and disease and their utility in managing the current COVID-19 pandemic.

**Keywords** biomarker; cell fitness; COVID-19; flower; prognosis
**Subject Categories** Biomarkers; Microbiology, Virology & Host Pathogen Interaction

## Introduction

To date, SARS-CoV-2 (the causative agent of COVID-19) has caused > 236 M infections and > 5 M deaths. While vaccines are now available, there are still approximately 500 K infections a day and COVID-19 will remain a public health problem for the foreseeable future. There is also a growing concern that areas with uncontrolled COVID-19 outbreaks will give rise to variants that evade vaccine-induced immunity. SARS-CoV-2 causes a broad spectrum of disease, ranging from asymptomatic to fatal infections (Flerlage *et al*, 2021). Essential in pandemic management is the development of prognostic

1–35 The list of affiliations appears at the end of this article
*Corresponding author. Tel: +41 612653229; E-mail: alexandar.tzankov@usb.ch
**Corresponding author. Tel: +351 210489380; E-mail: eduardo.moreno@research.fchampalimaud.org
***Corresponding author. Tel: +45 35331419 and +351 910225386; E-mail: rajangogna@gmail.com; rajan.gogna@bric.ku.dk
†These authors contributed equally to this work as first authors
‡These authors contributed equally to this work as second authors

biomarkers for COVID-19 patients, to help facilitate patient triage and resource prioritization. Specific patient demographics (e.g., older age) and comorbidities (e.g., diabetes, obesity, and/or cardiovascular disease) are associated with increased COVID-19 severity (Longmore et al, 2021). However, such clinical characteristics typically have limited prognostic value in COVID-19 patients (as demonstrated by an area under a curve [AUC] of < 0.75) (Yang et al, 2020a; Grifoni et al, 2021). Other prognostic biomarkers that measure the immune response to infection (e.g., C-reactive protein [CRP] and the neutrophil-lymphocyte ratio) have been proposed (Huang et al, 2020b). However, such biomarkers typically appear later in the infection and their utility depends on the early collection of patient blood, which may not be possible in all diagnostic settings. Rather, it would be ideal to develop a COVID-19 biomarker in nasopharyngeal samples, as this sample is almost always the first or the earliest diagnostic sample collected from patients with a suspected COVID-19 illness. Early evidence suggests nasopharyngeal samples can provide important prognostic information in the context of COVID-19 (Ziegler et al, 2021).

Markers of cellular fitness have yet to be investigated for their prognostic value in COVID-19. We and others have previously shown that tissues have an intrinsic surveillance mechanism by which cells continuously interact and sense the fitness status of their neighbors (Merino et al, 2016; Liu et al, 2019; Madan et al, 2019; Flanagan et al, 2021; van Neerven et al, 2021; Yum et al, 2021). Communication of cellular fitness is an essential mechanism through which relatively less fit or suboptimal cells are recognized and removed from tissues (Rhiner et al, 2010; Merino et al, 2013; Madan et al, 2019; Coelho & Moreno, 2020). We have identified a unique molecular mechanism of "fitness fingerprints", which executes this cell recognition and elimination system, thereby maintaining optimum tissue health over time (Rhiner et al, 2010; Merino et al, 2013). We recently found that this system functions through the human flower gene (hFwe) (Madan et al, 2019). In a number of human cancers, the noncancerous cells surrounding a tumor are forced to express a specific fitness mark, called flower lose (hFwe-Lose) (Madan et al, 2019). hFwe-Lose-expressing stromal cells are eliminated by competitive pressure from the aggressive, superfit, cancer cells (Moreno, 2008; Madan et al, 2019; Parker et al, 2020). This mechanism is also used by healthy tissues, where suboptimal cells, which arise due to external or internal influences (e.g., oxidative stress, inflammation, cytotoxicity, or radiation), express hFwe-Lose and are marked for elimination (Rhiner et al, 2010; Merino et al, 2016; Madan et al, 2019). Therefore, hFwe-Lose is a unique marker for suboptimal or unfit status of cells in many tissues or organ systems. Low division potential, genotoxic stress, cytotoxic stress, radiation, aging, and the generation of reactive oxygen species (ROS) negatively impact cellular fitness and may render a cell suboptimal (Wang et al, 2006; Bondar & Medzhitov, 2010; Rhiner et al, 2010; Merino et al, 2015; Akieda et al, 2019; Liu et al, 2019) (Fig 1A).

SARS-CoV-2 infection of alveolar epithelial cells can lead to cell death and tissue damage (Huang et al, 2020a; Xu et al, 2020). Specifically, the loss of epithelial cells in the lower respiratory tract disrupts the pulmonary epithelial–endothelial barrier, resulting in pulmonary edema and respiratory distress (Ackermann et al, 2020; Barton et al, 2020; Konopka et al, 2020; Li et al, 2020; Ren et al, 2020). SARS-CoV-2-induced cell death can also result in the release of inflammatory factors, which may further perpetuate the "cytokine storm"

associated with severe disease (Li et al, 2020). Cells under stress, which gain suboptimal status (i.e., those that are less fit), may be at an increased risk of virus-induced cell death. This stress can be associated with the chronic inflammation characteristic of different host comorbidities such as diabetes, obesity, and COPD (Milner & Beck, 2012; Bennett et al, 2018; Furman et al, 2019; Guan et al, 2020; Richardson et al, 2020; Yang et al, 2020b). We therefore propose that hFwe-Lose, a cell fitness biomarker, may be associated with a heightened risk of SARS-CoV-2-induced cell death and therefore could be an important prognostic biomarker in COVID-19 patients.

Here, we use samples from both the lower and upper respiratory tracts of COVID-19 patients to show that hFwe-Lose expression has a strong prognostic value for COVID-19 patient hospitalization and death. Indeed, hFwe-Lose expression in the upper respiratory tract outperformed other prognostic factors such as patient age, comorbidities, and clinical markers of inflammation in predicting patient outcome.

# Results

### hFwe-Lose expression in lung tissue increases with age and host comorbidities

To determine whether hFwe-Lose expression was elevated in the lungs of older adults, hFwe-Lose expression was analyzed by qPCR in 86 lung tissue biopsies taken from non-COVID-19 patients aged between 20 and 82 years. Older patients showed increased hFwe-Lose expression ($P < 6 \times 10^{-4}$) (Fig 1B). Specifically, patients over 70, as well as those aged between 60 and 70 years old, exhibited a significant upregulation of hFwe-Lose expression compared with patients younger than 60 years (Appendix Fig S1A). Besides older age, comorbidities such as hypertension, obesity, COPD, diabetes mellitus, and cardiovascular disease are known risk factors for severe COVID-19 (Guan et al, 2020; Richardson et al, 2020; Williamson et al, 2020; Yang et al, 2020b; Elezkurtaj et al, 2021; Gao et al, 2021). Consistent with a link between elevated hFwe-Lose expression and a risk of severe COVID-19, hFwe-Lose expression was elevated in lung tissue biopsies from non-COVID-19 patients with comorbidities (Fig 1C). hFwe-Lose expression was elevated in the lungs of patients with hypertension (HT; $n = 129$), obesity ($n = 45$), COPD ($n = 51$), diabetes ($n = 48$), cardiovascular disease (CVD; $n = 63$) versus disease-free control lungs ($n = 42$). hFwe-Lose expression increased with the total number of patient comorbidities (Appendix Fig S1B). hFwe-Lose expression is associated with the presence of "less fit" cells (Madan et al, 2019) which may be more prevalent in the organs of older adults and those with comorbidities, both of which are known risk factors for severe COVID-19 (Longmore et al, 2021). If hFwe-Lose expression in the respiratory tract has prognostic value in the context of COVID-19, one would hypothesize that patients who died of COVID-19 would display elevated hFwe-Lose expression. Figure 1D illustrates an increased expression of hFwe-Lose in lung tissue of patients with fatal COVID-19 ($n = 11$). Although these individuals had a number of comorbidities, hFwe-Lose expression was significantly higher in COVID-19 patients compared to non-COVID patients with comorbidities (Fig 1D). Moreover, among COVID-19 patients, hFwe-Lose expression inversely correlated with the number of days from first symptoms to

death ($P < 3 \times 10^{-11}$) (Appendix Fig S1C). This could suggest that *hFwe-Lose* expression may be useful in identifying, among at-risk patients (i.e., those with comorbidities), those individuals with an elevated risk of fatal COVID-19 outcome.

Collectively, the above findings suggest that *hFwe-Lose* expression may play a prognostic role in COVID-19 by identifying individuals in whom a large number of "unfit" cells are present and therefore are at increased risk of virus-induced cell death. To test this hypothesis, the lungs of 11 deceased COVID-19 patients were examined (Menter *et al*, 2020) (Appendix Table S1). Typically, the lungs of healthy patients are characterized by air-filled alveolar

spaces, thin alveolar septae/gas-exchange membrane, and the absence of inflammatory cells and congestion in pulmonary capillaries (Fig 1E1). In fatal COVID-19 patients' severe congestion, exudative diffuse alveolar damage (DAD) and thromboses were observed (Fig 1E2-5). Collapsing alveoli were also observed with epithelial detachment and micro-perforations as well as several cells with karyopyknosis and karyorrhectic figures (Fig 1E2-5). This is typically indicative of apoptosis of alveolar, endothelial, and interstitial cells in infected patients and includes hyaline membrane formation, erythrocyte extravasation, prominent, enlarged, often multinucleated, and often apoptotic, type II pneumocytes, interstitial

---

**Figure 1.  *hFwe-Lose* biomarker associated with COVID-19 mortality and host comorbidities.**

A  A schematic of cell competition process. Our bodies have a natural surveillance system that optimizes tissue fitness. The process of cell competition drives healthy tissues to force suboptimal, yet viable, loser cells to undergo cell death. Various stressors and insults cause cells to alter their properties and expression of fitness biomarkers. Cellular fitness comparisons lead to the elimination of loser cells that express *hFwe-Lose*, a biomarker of reduced fitness. This mechanism is responsible for actively restoring tissue homeostasis and has important implications in response to infections and the development of malignancies. ROS: reactive oxygen species.

B  *hFwe-Lose* mRNA expression is more abundant in elderly people. *hFwe-Lose* mRNA expression was analyzed by RT–qPCR in 86 lung tissue biopsies taken from non-COVID patients with age between 20 and 82 years. Older patients show a significant upregulation of *hFwe-Lose* expression. A log-linear regression model demonstrates a positive correlation between age and *hFwe-Lose* expression ($R^2 = 0.13$; slope confidence interval of 95% (CI) = [2.0–12.2]; *P*-value of the linear regression model $< 6 \times 10^{-4}$).

C  *hFwe-Lose* expression is elevated in lung tissue biopsies from patients with comorbidities. Box plot illustrates an increased expression of *hFwe-Lose* in lungs of patients with hypertension (HT; $n = 129$), obesity ($n = 45$), chronic obstructive pulmonary disease (COPD; $n = 51$), diabetes ($n = 48$), cardiovascular disease (CVD; $n = 63$) versus disease-free control lungs ($n = 42$). Patient`s age is depicted in color. Two-sided Student's *t*-test was performed for each comorbidity (compared to disease-free patients), and *P*-values are presented on the plot. The central band shows the median, the box indicates the interquartile range, and the whiskers extend to the most extreme points within the 1.5-fold distance of the interquartile range above and below the box.

D  *hFwe-Lose* expression is upregulated in lung tissue of COVID-19 patients. Box plot illustrates an increased expression of *hFwe-Lose* in lung tissue of patients diagnosed with COVID-19 ($n = 11$), individuals affected with host comorbidities ($n = 216$) versus disease-free control lungs ($n = 42$). Patient's age is depicted in color. Two-sided Student's *t*-test was performed (compared to disease-free patients), and *P*-values are presented on the plot. The central band shows the median, the box indicates the interquartile range, and the whiskers extend to the most extreme points within the 1.5-fold distance of the interquartile range above and below the box.

E  SARS-CoV-2 infection manifests histological changes in patients' lungs as demonstrated by H&E staining. 1) For comparison, the normal lung of an elderly individual is shown, containing air-filled empty looking alveolar spaces and thin alveolar septae/gas-exchange membranes. Note that there are almost no inflammatory cells and that capillaries are only merely visible since not congested. 2) Diffuse (proliferative) alveolar damage showing evidence of cell death. An alveolar space filled with desquamated pneumocytes and macrophages, lymphocytes, and focal erythrocyte extravasation as well as one multinucleated pneumocyte type II. The still recognizable epithelial lining is detached. Cells and surfactant are lost creating perforations in the alveolar wall, allowing migration of blood cells and fluid to enter inside the alveolar space. The adjacent interstitial space is significantly widened, showing an increase in mononuclear inflammatory cells and extravasation of erythrocytes. On the top and the lower right, dilated and congested capillaries can be seen (hematoxylin and eosin (H&E) stain, 200×). a—alveolar structure destroyed by lymphocytes, desquamated epithelium, and extravasated blood; the alveoli should be empty, but here it is filled with a combination of degenerated cells and fibrin; b—next alveoli; c—expanded pulmonary interstitium. The interstitium should be as thin as 10 µm, but here it is 100 µm; d—extravasation of erythrocytes; e—lymphocytes; f—congested capillaries and arterioles; g—multinucleated alveolocyte/pneumocyte type II; h—detached epithelium; i—a histiocyte with a kidney-shaped nucleus at the bottom of the circle has ingested erythrocytes (left-handed), representing the first step of cellular elimination. Round inlet with encircled karyopyknotic, karyorrhectic, and "ghost-cell" figures indicative of apoptosis. 3) The presence of type II pneumocyte syncytial giant cells in a collapsing alveolar space with detached epithelial lining. The adjacent interstitial space shows analogous changes to the previous example. At the bottom, there is a prominently dilated and congested capillary (H&E, 200×). a—alveoli with detached epithelium; b—next alveoli; c- Extravasation of erythrocytes; d—expanded interstitium; e—lymphocytes; f—multinucleated pneumocyte type II. 4) Diffuse alveolar damage showing massive extravasation of fibrin (homogeneous eosinophilic material in the center of the alveolar space). The lining pneumocytes are almost all apoptotic/necrotic. The fibrin exudate is intermingled with mononuclear inflammatory cells and cellular debris (H&E, 200×). a—alveola; b—congested capillaries; c—fibrin; d—remnants of degenerated/dying epithelium; e—expanded interstitium; f—cellular debris consisting of macrophages, detached epithelial cells and lymphocytes; g—fully degenerated/lacking epithelial coverage within the alveoli. Round inlet with encircled karyopyknotic, karyorrhectic, and "ghost-cell" figures indicative of apoptosis. 5) Immunohistochemical stain (IHC) for fibrin showing microthrombi caused by dysfunction of endothelial cells in capillaries of the alveolar membranes/lung interstitium leading to obstruction of the microcirculation (IHC for fibrin, 200×). a—all small alveolar septal capillaries are filled out with worm-like fibrin thrombi hampering / obstructing circulation; b—normal interstitium.

F  Healthy lungs show very low cleaved caspase-3-positive cells. IHC staining for cleaved caspase-3 (20×).

G  Expression of cleaved (active) caspase-3 yielding apoptotic and pre-apoptotic cells: IHC for cleaved (active) caspase-3 showing a brown nuclear staining signal in respective cells. 31.5% cells were found to be caspase positive, 11.7% with high caspase-3 positivity and another 19.8% with low. These cells are mainly located in the interstitium, but also in some alveolar epithelial and endothelial compartments. In the lower right, the edge of an alveolar space containing several (pre-)apoptotic pneumocytes is seen, while in the upper left, a completely denuded/deepithelialized alveolus with two apoptotic remnants is observable (IHC for cleaved caspase-3, 20×). a—apoptotic large mononuclear cells; b—apoptotic alveolocyte / pneumocyte; i—amplification of apoptotic alveolocyte / pneumocyte. As shown in the round inlets of 1E2 and 1E4, homogeneous dark nuclear condensation up to a size of 2 µm (karyopyknosis), homogeneous pinkish-grayish nuclear condensation, large nuclear inclusion with marginalization of the condensed chromatin, coarse nuclear angulation, nose-like/polar body-like nuclear protrusions, the latter two being karyorrhectic debris, were all morphologically considered evidence of apoptosis. This has been correlated with and was reflected by the results of the cleaved caspase-3 staining on step sections. All cells were counted on the 20× fields for $n = 42$ disease-free and $n = 3$ COVID-19 autopsy patients. The expression of *hFwe-Lose* in COVID-19 patients is significantly higher than in disease-free individuals, regardless of cleaved caspase-3 staining (Cas POS: $P < 0.004$; Cas NEG: $P < 0.006$). In COVID-19 patients, we observed a significantly higher expression of *hFwe-Lose* in sections with positive cleaved caspase-3 staining ($P < 0.004$). Two-sided Student's *t*-test was performed (compared to disease-free samples and NEG samples, respectively), and *P*-values are presented on the plot. The central band shows the median, the box indicates the interquartile range, and the whiskers extend to the most extreme points within the 1.5-fold distance of the interquartile range above and below the box.

---

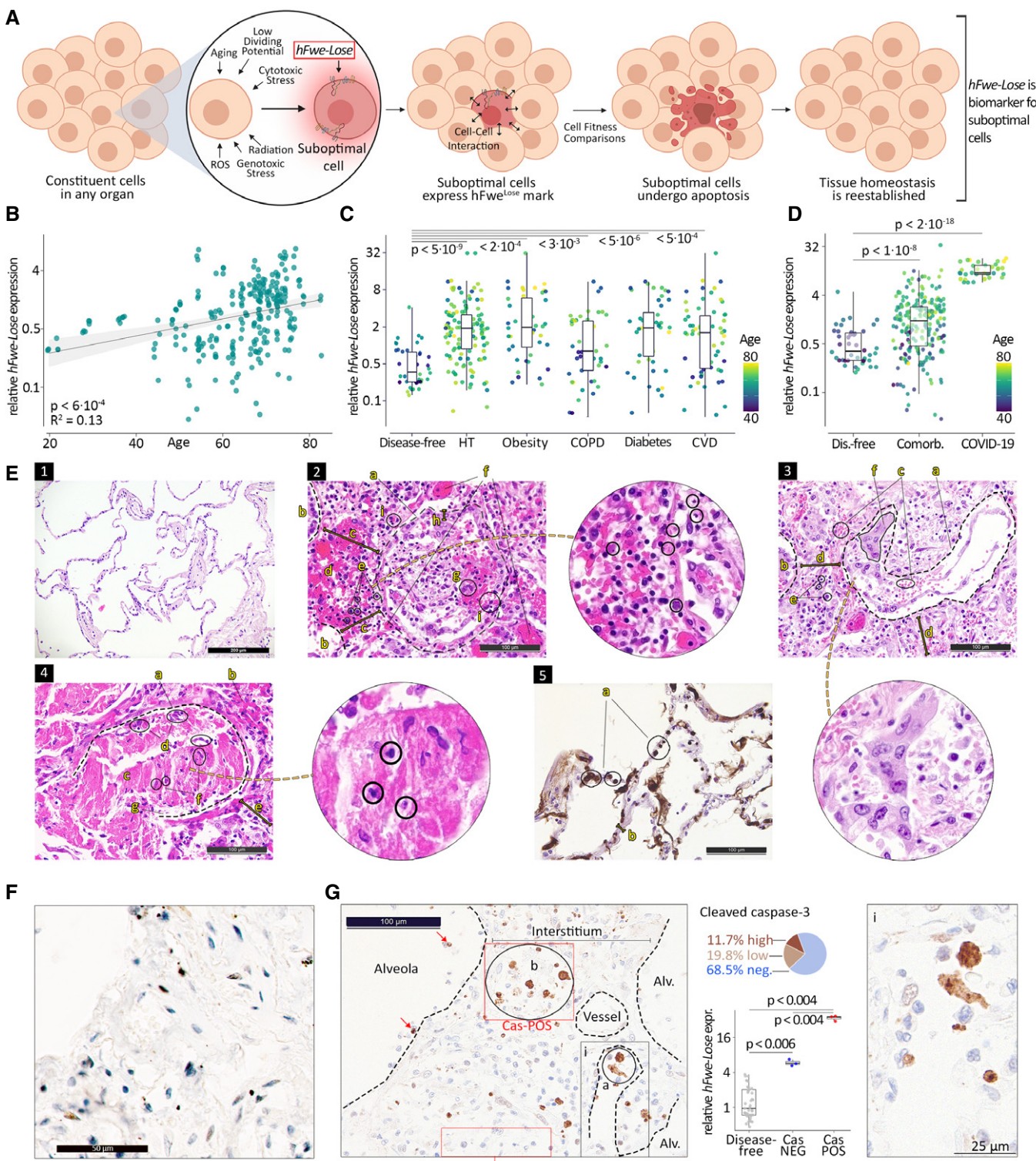

Figure 1.

expansion/edema with increased lymphocytes, and histiocytes (Fig 1E2-5). To further investigate cell death in these lungs, and the association with *hFwe-Lose* expression, sections from control and COVID-19 patients were stained for active/cleaved caspase-3.

Limited caspase staining was observed in disease-free lungs (Fig 1 F). Interestingly, in COVID-19 patients the caspase staining was present in "patches", with the presence of both caspase-positive and caspase-negative regions (Fig 1G). This is suggestive of clonal

expansion of cells with markedly different vulnerability to apoptosis within distinct niches, as is expected in the case of flower fitness-based cellular selection and subsequent expansion. To this end, we laser-captured such regions and examined the expression of *hFwe-Lose*. We found that the caspase-positive regions had significantly higher expression, compared to disease-free lung samples (Fig 1G, right). These results indicate that high expression of the suboptimal fitness marker *hFwe-Lose* correlates with areas of increased cell death in the lung ($P < 0.004$).

### *hFwe-Lose* expression in nasopharyngeal swabs is associated with poor outcomes in COVID-19 patients

While the above data suggest that *hFwe-Lose* can be clearly detected in the lower respiratory tract in individuals at risk of severe COVID-19, as well as deceased COVID-19 patients, the lower respiratory tract is not well-suited for the detection of prognostic biomarkers, due to the difficulties associated with obtaining samples. In contrast, nasopharyngeal swabs are routinely performed in COVID-19 patients and represent a more accessible clinical sample for the discovery of prognostic biomarkers. Accordingly, nasal swabs were obtained from COVID-19 patients ($n = 283$) aged between 1 and 96 years. Swabs were taken at the very beginning of disease onset (i.e., at earliest contact with a physician, before the disease progression). The samples were taken at two independent centers from cohorts with similar baseline characteristics (Table 1). For prognostic testing purposes, we split the entire dataset into a retrospective cohort (training, $n = 203$) and a prospective cohort (validation, $n = 80$) that were used to train and validate the predictive models. Consistent with our lower respiratory tract data, *hFwe-Lose* expression in the upper respiratory tract increased with the age of COVID-19 patients (Fig 2A). The positive association of *hFwe-Lose* expression and age is also seen when analyzing patients without associated comorbidities or patients with a single comorbidity (Appendix Fig S2A). At all ages, *hFwe-Lose* expression was lower in the patients that were not hospitalized (Fig 2A). *hFwe-Lose* expression was also elevated in the nasopharyngeal swabs of patients with comorbidities associated with severe COVID-19, namely diabetes mellitus ($n = 129$), COPD ($n = 20$), obesity (BMI > 30; $n = 152$), cardiomyopathy (CM; $n = 19$), heart failure (HF; $n = 35$), hypertension (HT; $n = 121$), and chronic kidney disease (CKD; $n = 60$), compared to disease-free patients (Fig 2B). Notably, many patients had multiple comorbidities at the same time (disease-free = 96, one comorbidity = 76, two comorbidities = 55, three comorbidities = 33, four comorbidities = 18, five comorbidities = 5) (Appendix Fig S2B). Since *hFwe-Lose* expression increases with age, and older adults are more inclined to have comorbidities (Niccoli & Partridge, 2012), we created an age-adjusted statistical model to determine an unbiased correlation of comorbidity status and *hFwe-Lose* expression. The adjusted model suggests that hypertension ($P < 9 \times 10^{-9}$) and diabetes ($P < 0.04$) have the highest age-independent effect on *hFwe-Lose* expression level (Fig 2C). Importantly, among COVID-19 patients who were hospitalized within 14 days of disease progression ($n = 177$), nasopharyngeal *hFwe-Lose* expression was elevated in patients who: were admitted to ICU ($n = 34$), underwent intubation ($n = 58$), were in respiratory distress (defined as an elevated respiratory rate greater than 30) ($n = 76$), had blood oxygenation level (SpO2) less than 94% ($n = 147$), and died within 30 days of

disease progression ($n = 21$) (Fig 2D). The expression of *hFwe-Lose* was also higher in patients who were hospitalized within 14 days of disease progression ($n = 177$) and who died within 30 days of disease progression ($n = 21$) versus patients who were not hospitalized (Fig 2E). We fitted a logistic model to predict the probability of hospitalization or death based on *hFwe-Lose* expression in nasal swab samples (Fig 2F). The logistic model was adjusted for age and sex. This model predicts a > 50% chance of hospitalization for (otherwise still healthy, not yet suffering from COVID-19) people who have a *hFwe-Lose* expression in their nasal swab samples that is approximately twice as large as the average *hFwe-Lose* expression in the group of non-hospitalized patients ($P < 0.001$).

### *hFwe-Lose* expression in nasopharyngeal swabs predicts outcome in COVID-19 patients

As a complementary approach to the logistic model, we used classification and regression tree (CART) analysis to find which of the variables (*hFwe-Lose* expression, age, sex, blood biomarkers, and presence of comorbidities) are valuable for the prognosis of the outcome. This method tries to find a series of partitions of the dataset that provide the most information about the outcome. This results in a tree-like structure of decisions based on the values of the variables used for prediction. The tree based on all patients ($n = 283$) is shown in Fig 3A. The partition providing the most information about the outcome is visualized as a split and is based on the *hFwe-Lose* expression. This split is done on all patients, whose outcomes are denoted underneath each split (gray: non-hospitalized, blue: hospitalized, red: dead). The subgroup of patients with *hFwe-Lose* expression < 2.45 are then split by age, and the subgroup with higher *hFwe-Lose* expression is once again split by *hFwe-Lose* expression, separating those patients with very high expression (> 4.41) from those with moderate to high expression (2.45–4.41). With each split, the coefficient of determination increases, and the relative error decreases (Barlin *et al*, 2013). The first split (*hFwe-Lose* > 2.45) increased the coefficient of determination by ~45% and reduced the relative error by ~45%. The impact of all following splits on the relative error was marginal (Fig 3B). The CART suggests the following classification of patients based on *hFwe-Lose* expression and age: (i) *hFwe-Lose* > 4.41 and age > 75 results in prediction "death". (ii) *hFwe-Lose* > 4.41 and age < 75 results in prediction "hospitalization". (iii) *hFwe-Lose* between 2.45 and 4.41 results in prediction "hospitalization". (iv) *hFwe-Lose* < 2.45 and age < 15 results in prediction "hospitalization". (v) *hFwe-Lose* between 1.1 and 2.45 and age > 44 results in prediction "hospitalization". (vi) *hFwe-Lose* between 1.1 and 2.45 and age < 44 results in prediction "non-hospitalized". (vii) *hFwe-Lose* < 1.1 results in prediction "non-hospitalized". Together, these data show that *hFwe-Lose* expression and patient age were the most useful predictors of COVID-19 outcome ($P < 0.01$). Figure 3C shows the impact of each possible predictor on the accuracy or misclassification rate that is measured as the mean decrease in the Gini coefficient obtained for the respective factor in a random forest analysis. *hFwe-Lose* expression had the highest score, followed by age and blood biomarkers. The presence of comorbidities obtained the lowest scores.

We next analyzed the impact of known COVID-19 blood biomarkers (ferritin, CRP, D-dimer, and neutrophil-lymphocyte ratio) and *hFwe-Lose* expression in nasopharyngeal swabs on the

**Table 1.** Patient data: Retrospective (training) and prospective (validation) cohorts.

| Characteristics | All (N = 283) | Retrospective cohort (N = 203) | Prospective cohort (N = 80) |
|---|---|---|---|
| Age, Q2 [Q1, Q3], (range), years | 55 [41,67] (1–96) | 54 [41.5,66] (1–96) | 55 [39.5,68] (5–91) |
| Male, no. (%) | 50.4 | 47.3 | 54.5 |
| Active smoker (%) | 9 (3%) | 5 (2%) | 4 (5%) |
| Former smoker (%) | 82 (29%) | 59 (29%) | 23 (29%) |
| BMI, Q2 [Q1, Q3], (range) | 28.3[24.7,34] (14.7–81.2) | 28.4 [24.8,34] (15.2–81.2) | 27.9 [24.3,34] (14.7–50.9) |
| Comorbidities, no. (%) | | | |
| Cancer | 102 (36%) | 67 (33%) | 35 (44%) |
| Chronic kidney disease | 35 (21%) | 35 (17%) | 25 (31%) |
| COPD | 20 (7%) | 12 (6%) | 8 (10%) |
| Heart failure | 35 (12%) | 15 (7%) | 20 (25%) |
| Cardiomyopathy | 19 (7%) | 10 (5%) | 9 (11%) |
| Solid organ transplant | 18 (6%) | 9 (4%) | 9 (11%) |
| Sickle cell disease | 5 (2%) | 2 (1%) | 3 (4%) |
| Type 2 diabetes mellitus | 79 (28%) | 48 (24%) | 31 (39%) |
| Hypertension | 121 (42%) | 95 (47%) | 26 (33%) |
| Medical care, no. (%) | | | |
| Physicians' evaluation as severe | 68 (24%) | 35 (17%) | 33 (41%) |
| Physicians' evaluation as moderate | 79 (28%) | 59 (29%) | 20 (25%) |
| Physicians' evaluation as mild | 136 (48%) | 109 (54%) | 27 (34%) |
| Intubation (%) | 58 (20%) | 33 (16%) | 25 (31%) |
| SpO2 < 94% | 147 (52%) | 94 (46%) | 53 (66%) |
| Respiration GT30 | 76 (27%) | 41 (20%) | 35 (44%) |
| Remdesivir treatment | 62 (22%) | 32 (16%) | 30 (38%) |
| Length of stay in hospital Q2 [Q1, Q3], (range), days | 7.1 [3.8,14.2] (1–99) | 6.9 [3.6,11.8] (1–99) | 8.3 [4,17.9] (1–40) |
| Outcome | | | |
| Hospitalized within 14 days (%) | 177 (63%) | 116 (57%) | 61 (76%) |
| Death within 30 days | 21 (7%) | 10 (5%) | 11 (14%) |
| Blood tests | | | |
| Neutrophils, per µl Q2 [Q1, Q3], (range), (norm: 200–1000) | 4940 [3000,8190] (100–30860) | 4675 [2990,6975] (100–19640) | 6930 [3810,10360] (1430–30860) |
| Lymphocytes (range), per µL Q2 [Q1, Q3], (range), (norm: 100–300) | 940 [700,1240] (50–2910) | 980 [730,1308] (220–2910) | 730 [570,1020] (50–1730) |
| Neutrophils/Lymphocytes ratio Q2 [Q1, Q3], (range), (norm < 6.5) | 5.1 [3.1,9.2] (0.2–98.3) | 4.8 [2.9,7.6] (0.2–45.7) | 8.4 [4.4,16.5] (1.6–98.3) |
| Ferritin Q2 [Q1, Q3], (range), ng/ml (norm: 22–322; high: 322–900; very high: > 900) D–dimer Q2 [Q1, Q3], (range), µg FEU/ml | 608 [222,1599] (20–7518) | 494 [197,1457] (20–7518) | 836 [319,1878] (89–3614) |
| D–dimer Q2 [Q1,Q3], (range), µg FEU/ml (norm: < 0.5; high: 0.5–1; very high: > 1) | 1.3 [0.8,2.7] (0.2–88.8) | 1.1 [0.7,2.4] (0.2–88.8) | 1.7 ± [1,3.6] (0.5–20.0) |
| CRP Q2 [Q1, Q3], (range), mg/dl (norm: < 5; high: 5–50; very high: > 50) | 5.9 [2.1,13.9] (0.1–35.8) | 5.4 [1.5,12.1] (0.1–33.6) | 6.8 [2.8,17] (0.4–35.8) |
| hFwe-Lose Q2 (Q1, Q3), (range), relative expression | 2.7 [1.1,3.9] (0.1–5.5) | 2.7 [1,4.0] (0.1–5.5) | 2.7 [1.7,3.6] (0.4–4.8) |

Clinical characteristics of the study patients (n = 283 total patients) for Figs 2–4, Appendix Figs S1 and S2. Patient Characteristics, comorbidities, clinical evaluation, hospitalization status, blood findings and hFwe-Lose measurements have been tabulated in the retrospective (n = 203) and prospective (n = 80) arms of the study.

age- and sex-adjusted probability of the possible outcomes (non-hospitalized, hospitalized, or dead) using mutlinominal logistic models (Appendix Fig S2C, upper row). A statistically significant association was detected between COVID-19 outcome and the concentration of all biomarkers, with an increasing probability of worse outcomes associated with increasing biomarker concentrations. However, the predictive value of these biomarkers was low, due to the large variance observed among hospitalized patients. Only *hFwe-Lose* expression was able to distinguish non-hospitalized and hospitalized outcomes. Patients with a relative *hFwe-Lose* expression of < 1.5 would be classified as remaining non-hospitalized, those with a larger relative expression would be classified as becoming hospitalized, whereas all other biomarkers would classify all patients as becoming hospitalized. None of the models were able to classify death as the outcome.

We additionally used binary logistic models to predict the age- and sex-adjusted probability of death from the same biomarkes (Appendix Fig S2C, lower row). All models showed a positive

association of probability of death with increasing concentrations of the biomarker, but here the association between death and D-dimer concentration was not statistically significant. These models can be compared by the Akaike information criterion (AIC) that estimates the relative amount of information that the biomarker provides about the outcome classification. Lower values indicate a higher relative information content. The AIC values of all blood biomarkers are comparable, whereas the AIC for the *hFwe-Lose* expression is considerably lower, indicating that *hFwe-Lose* expression provides the most useful information to judge whether a COVID-19 patient may eventually succumb to the infection.

To further evaluate the prognostic capacity of *hFwe-Lose* expression, prognostic receiver operator characteristic value (ROC) curves were generated using the retrospective training cohort of 203 patients. Using ROC curve analysis, we identified the prognostic cut-off value of *hFwe-Lose* expression > 3.17 (for hospitalization; TPR = 0.77, FPR = 0.03) and *hFwe-Lose* expression > 4.44 (for death; TPR = 0.1, FPR = 0.08). The associated AUC values were 0.98 and

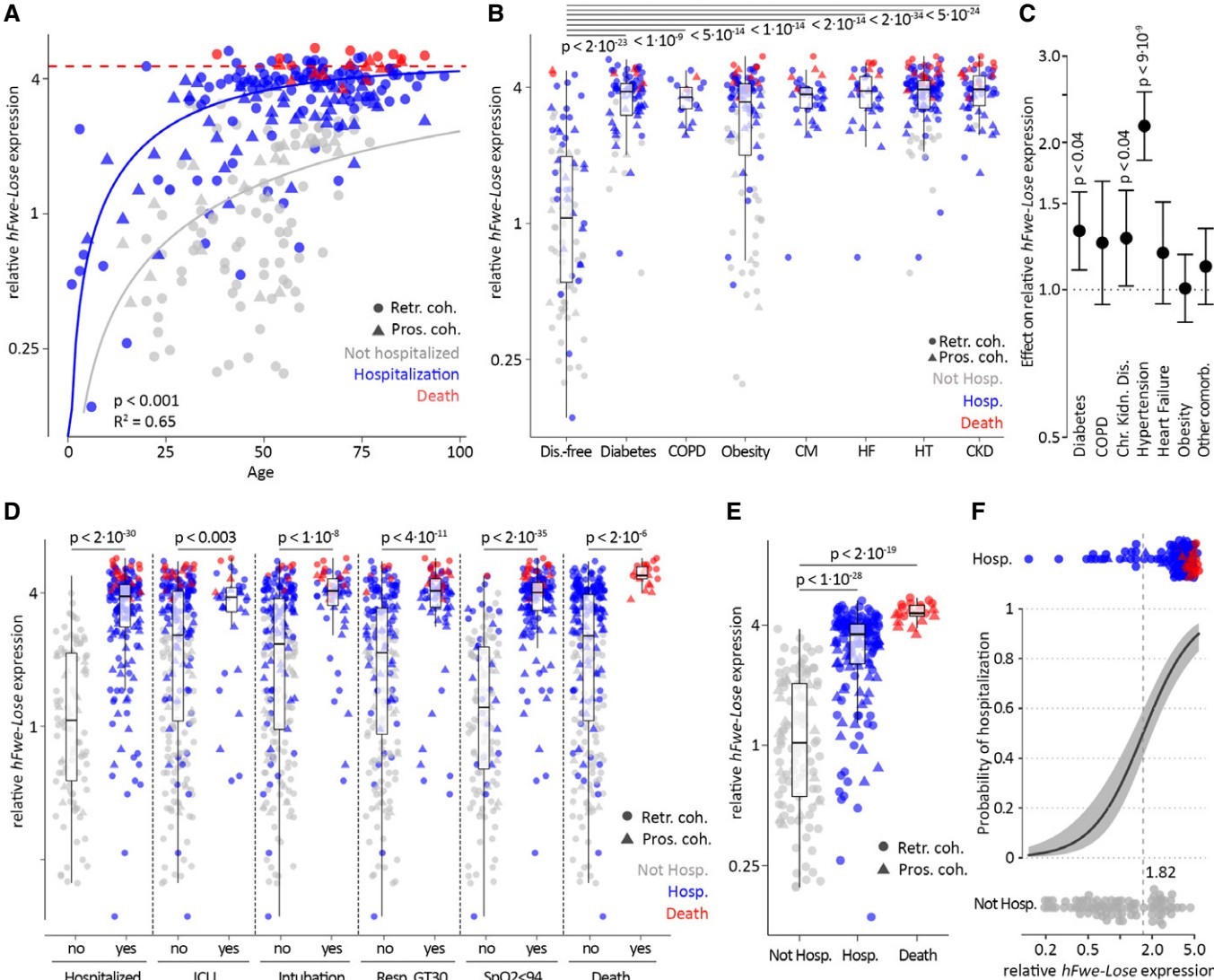

**Figure 2.**

**Figure 2. *hFwe-Lose* biomarker, measured in nasopharyngeal swab samples, associates with patients' COVID-19 disease outcome.**

A   *hFwe-Lose* biomarker expression is more abundant in nasopharyngeal swab probes from older adults. *hFwe-Lose* expression was analyzed by RT–qPCR in 283 nasopharyngeal swab samples taken from patients with age between 1 and 96 years, taken at the very beginning of the disease (the earliest contact with physician, before the disease progression). The vertical axis represents relative *hFwe-Lose* expression normalized to the mean of non-hospitalized patients. Colors depict the outcome groups: non-hospitalized (gray, $n = 85$), hospitalized (blue, $n = 177$), and deceased (red, $n = 21$). The shape of data points reflects the cohorts: circles for the training cohort ($n = 203$) and triangles for the validation cohort ($n = 80$). The lines show the fitted curves of an asymptotic model with the same asymptotic value but different rate constants per group (see Materials and Methods). Due to the comparatively low number of deceased patients in the dataset ($n = 21$), the curve for this group reflects the asymptotic value. Hospitalized and deceased patients show a positive correlation of *hFwe-Lose* expression and age with a larger rate constant for the hospitalized patients ($R^2 = 0.65$). The *P*-value (< 0.001) indicates that the blue curve (for hospitalized patients) grows faster with age, compared to the gray curve (for non-hospitalized patients).

B   *hFwe-Lose* expression is elevated in nasopharyngeal swab probes from patients with comorbidities. Box plots illustrate an increased relative expression of *hFwe-Lose* in nasopharyngeal swabs of patients with diabetes ($n = 129$), COPD ($n = 20$), obesity (BMI > 30; $n = 152$), cardiomyopathy (CM; $n = 19$), heart failure (HF; $n = 35$), hypertension (HT; $n = 121$), chronic kidney disease (CKD; $n = 60$) versus disease-free patients ($n = 96$). Two-sided Student's *t*-tests were performed (compared to disease-free patients), and *P*-values are presented on the plot. The vertical axis represents relative *hFwe-Lose* expression normalized to the mean of non-hospitalized patients. The color refers to the COVID-19 disease outcome: gray for not hospitalized, blue for hospitalized and red for deceased patients. The shape of data points reflects the cohorts: circles for the training cohort ($n = 203$) and triangles for the validation cohort ($n = 80$). The central band shows the median, the box indicates the interquartile range, and the whiskers extend to the most extreme points within the 1.5-fold distance of the interquartile range above and below the box.

C   An age- and sex-adjusted statistical model suggests hypertension, diabetes, and chronic kidney disease to have the highest impact on *hFwe-Lose* expression. A linear regression model was created to account the patient's age upon analysis of *hFwe-Lose* expression in relation to comorbidity status. "Other comorbidity" refers to a cumulative effect of diseases or conditions, not directly associated with COVID-19 (cancer, Down syndrome, solid organ transplant, sickle cell disease, bone marrow transplant). The plot illustrates the effect of selected comorbidities (horizontal axis) on relative *hFwe-Lose* expression (vertical axis). The *P*-values of the respective linear models for all significant comorbidities are presented on the plot. The error bars represent the 95% confidence interval.

D   Elevated *hFwe-Lose* expression in the nasal swab samples associates with patients' condition severity and respective medical treatment. Box plots illustrate an increased expression of *hFwe-Lose* in nasal swabs of patients, who were hospitalized within 14 days of disease progression ($n = 177$), admitted to intensive care unit (ICU) ($n = 34$), underwent intubation ($n = 58$), had respiratory rate greater than 30 (GT30; $n = 76$), had blood oxygenation level (SpO2) less than 94% ($n = 147$), and died within 30 days of disease progression ($n = 21$) versus patients without respective conditions. Pairwise two-sided Student's *t*-tests were performed (compared to patients without respective conditions), and *P*-values are presented on the plot. The vertical axis represents *hFwe-Lose* expression normalized to the mean of non-hospitalized patients. The color refers to the COVID-19 disease outcome: gray for non-hospitalized, blue for hospitalized, and red for deceased patients. The shape of data points reflects the cohorts: circles for the training cohort ($n = 203$) and triangles for the validation cohort ($n = 80$). The central band shows the median, the box indicates the interquartile range, and the whiskers extend to the most extreme points within the 1.5-fold distance of the interquartile range above and below the box.

E   Elevated *hFwe-Lose* expression in nasal swab associates with patients' disease outcome. Box plot emphasizes an increased expression of *hFwe-Lose* in nasal swabs of patients, who were hospitalized within 14 days of disease progression ($n = 177$), and who died within 30 days of disease progression ($n = 21$) versus patients without respective conditions. Two-sided Student's *t*-tests were performed (compared to non-hospitalized patients), and *P*-values are presented on the plot. The vertical axis represents *hFwe-Lose* expression normalized to the mean of non-hospitalized patients. The color refers to the COVID-19 disease outcome: gray for not hospitalized, blue for hospitalized, and red for deceased patients. The shape of data points reflects the cohorts: circles for the training cohort ($n = 203$) and triangles for the validation cohort ($n = 80$). The central band shows the median, the box indicates the interquartile range, and the whiskers extend to the most extreme points within the 1.5-fold distance of the interquartile range above and below the box.

F   The logistic model predicts probability of hospitalization based on *hFwe-Lose* expression in nasal swab samples. This model predicts a > 50% chance of hospitalization for people (otherwise still healthy, not yet infected with COVID-19), who have a *hFwe-Lose* expression in their nasal swab samples >1.82 than the mean of non-hospitalized patients. *P*-value of the logistic model < 0.001. The gray area shows the 95% confidence band.

0.89 for hospitalization and death, respectively (Fig 3D). Consistent with these data, there was increased expression of *hFwe-Lose* in nasal swabs of patients, who were hospitalized or died versus patients who were not hospitalized for both the retrospective (training; $n = 203$) and prospective (validation; $n = 80$) patient cohorts (Fig 3E). In the retrospective cohort, the outcome forecast was 84% correct in predicting non-hospitalization, 63% correct for hospitalization, and 100% correct for death. In the prospective cohort, the outcome prediction was 100% correct for non-hospitalization, 72% correct for hospitalization, and 55% correct for death (45% of deceased patients were predicted to be "only" hospitalized; none of deceased patients had a "not hospitalized" prediction) (Fig 3E). Confusion matrices and heatmaps were next used to visualize the sensitivity and specificity of selected cut-offs (*hFwe-Lose* expression > 3.17 for hospitalization and *hFwe-Lose* expression > 4.44 for death) (Fig 3F). In the retrospective cohort, 72 out of the total 86 not hospitalized patients were correctly predicted. Out of 107 hospitalized patients, 67 were correctly predicted while 19 were predicted to die instead and 21 were not predicted to be hospitalized. All deceased patients were correctly predicted. In the prospective cohort, all patients (19) who were not hospitalized, were correctly

predicted. Out of 50 hospitalized patients, 36 were correctly predicted while 14 were predicted to be non-hospitalized. Finally, out of 10 deceased patients in the prospective cohort, 6 were correctly predicted and 5 patients were predicted to be "only" hospitalized (Fig 3F). For hospitalization prediction, positive predictive value (PPV) for the retrospective cohort was 83.7 and 87.8% for the prospective cohort. The negative predictive value (NPV) for the retrospective cohort was 67.2 and 64.1% for the prospective cohort. For death prediction, the PPV for the retrospective cohort was 34.5% while it was 100% for the prospective cohort. The NPV for the retrospective cohort was 100 and 93.2% for the prospective cohort (Fig 3F). These high negative predictive values make *hFwe-Lose* expression as a very useful biomarker for assessing mortality risk (i.e., to exclude the risk of COVID-19-related deaths), thereby confirming its potential role in triage and risk stratification of COVID-19 patients.

Finally, we evaluated the additional prognostic value of nasopharyngeal *hFwe-Lose* expression over and above conventional methods routinely used in clinical settings, such as laboratory tests alone (e.g., blood-based biomarkers) or laboratory tests together with clinical information (e.g., age and comorbidities). Specifically, the AUC

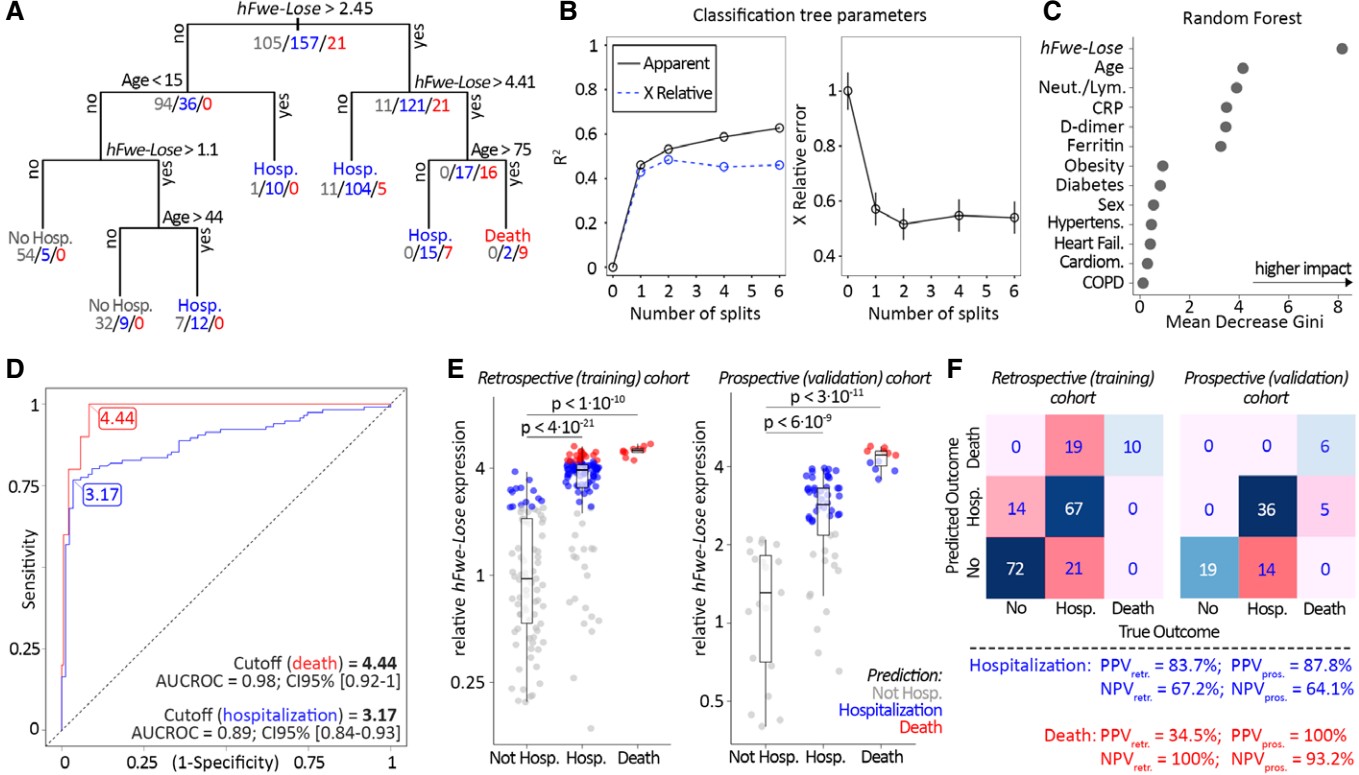

**Figure 3. hFwe-Lose biomarker, measured in nasal swab samples, predicts patients' COVID-19 outcome.**

A  Classification and regression tree (CART) shows that hFwe-Lose expression in patients' nasal swab sample and patients' age is the main predictors of the outcome. The classification tree was generated using all relevant information about patients (hFwe-Lose expression, age, sex, presence of comorbidities (diabetes, COPD, obesity, cardiomyopathy, heart failure, hypertension)). All patients were included in the CART analysis (n = 283). The CART algorithm selected hFwe-Lose expression and age as sole factors to determine the patients' outcome. The split cut-offs, which produce tree branches, are aimed to maximize the information gain (decrease of entropy) with each split, and in this way, the coefficient of determination increases, and the relative error decreases with each split (Barlin et al, 2013).

B  The line plots show saturation of the coefficient of determination, as well as the plateau in X relative error. P-value of the classification and regression tree analysis (for all splits) < 0.01. The first split (hFwe-Lose > 2.45) increased the coefficient of determination by ~45% and reduced the relative error by ~45%. The impact of all following splits on the relative error was irrelevant. The error bars (plot on the right) represent ± SE.

C  The random forest analysis shows the highest impact of the hFwe-Lose biomarker in the multivariate analysis of outcome prediction. The plot shows the Mean Decrease in Gini coefficients for the factors that were incorporated in the statistical model for the multivariate CART analysis. The Gini coefficient is a measure of the misclassification rate. The importance of a predictor is assessed by how much the predictor reduced this misclassification rate. The hFwe-Lose expression has the highest score, followed by age and blood biomarkers. Comorbidities show the least impact on reducing the misclassification rate.

D  hFwe-Lose is a sensitive and specific biomarker that predicts poor COVID-19 outcome. The ROC curves illustrate the high sensitivity and (1 - specificity) of FC > 3.17 (for hospitalization; TPR = 0.77, FPR = 0.03) and FC > 4.44 (for death; TPR = 0.1, FPR = 0.08) threshold levels (AUC = 0.89 and 0.98, respectively) in prediction patients' hospitalization and death. CI 95% (hospitalization) - [0.84–0.93]; CI (death) - [0.92–1]. Only retrospective (training, n = 203) patients' cohort was used for the creation of ROC curves.

E  Elevated hFwe-Lose expression in nasal swab probes predicts patients' disease outcome. Box plots show an increased expression of hFwe-Lose in nasal swabs of patients, who were hospitalized or died, versus patients who were not hospitalized, for retrospective (training; n = 203) and prospective (validation; n = 80) patients' cohorts. Two-sided Student's t-test was performed, and P-values are presented on the plot. The vertical axis represents hFwe-Lose expression normalized to the mean of non-hospitalized patients. The color refers to the COVID-19 disease outcome prediction: gray for not hospitalized, blue for hospitalized, and red for deceased patients. In the retrospective cohort, the outcome prediction was 84% correct in predicting non-hospitalization, 63% correct in prediction of hospitalization, and 100% correct in death prediction. In the prospective cohort, the outcome prediction was 100% correct in predicting non-hospitalization, 72% correct in prediction of hospitalization, and 55% correct in death prediction (45% of deceased patients were predicted to be "only" hospitalized; none of deceased patients had "not hospitalized" prediction). Two-sided Student's t-tests were performed (compared to non-hospitalized patients), and P-values are presented on the plot. The central band shows the median, the box indicates the interquartile range, and the whiskers extend to the most extreme points within the 1.5-fold distance of the interquartile range above and below the box.

F  The confusion matrices and heatmaps visualize the classification performance of hFwe-Lose expression at the selected cut-offs (FC > 3.17 for hospitalization and FC > 4.44 for death). In the retrospective (training) cohort, 72 out of the total 86 not hospitalized patients were correctly predicted; out of 107 hospitalized patients, 67 were correctly predicted, 19 were predicted to die instead, and 21 were not predicted to be hospitalized; all deceased patients were correctly predicted. In the prospective (validation) cohort, all patients (19) who were not hospitalized, were correctly predicted; out of 50 hospitalized patients, 36 were correctly predicted, 14 were predicted to be non-hospitalized; out of 10 deceased patients, 6 were correctly predicted, and 5 patients were predicted to be "only" hospitalized. For hospitalization prediction, positive predictive value (PPV) for retrospective (training) cohort is 83.7% and for prospective (validation) cohort is 87.8%. The negative predictive value (NPV) for retrospective (training) cohort is 67.2% and for prospective (validation) cohort is 64.1%. For death prediction, positive predictive value (PPV) for retrospective (training) cohort is 34.5% and for prospective (validation) cohort is 100%. The negative predictive value (NPV) for retrospective (training) cohort is 100% and for prospective (validation) cohort is 93.2%.

of nasopharyngeal *hFwe-Lose* expression was compared with blood-based biomarkers and/or patient comorbidities and age. The comorbidities used to train models included cancer, chronic kidney disease, COPD, down syndrome, heart failure, cardiomyopathy, solid organ transplant, sickle cell disease, type 2 diabetes mellitus, bone marrow transplant, and hypertension. Firstly, *hFwe-Lose* expression was compared with four known blood biomarkers (serum ferritin, CRP, D-dimer, and neutrophil-lymphocyte ratio) in predicting death of hospitalized COVID-19 patients (Fig 4A). For each biomarker, only patients who had laboratory testing performed (on admission to hospitals) were used. With AUC as the criteria, *hFwe-Lose* significantly outperformed all four biomarkers in predicting death (Fig 4A). Machine learning analysis further demonstrated that *hFwe-Lose* biomarker expression was superior to using patient age and comorbidities to predict poor COVID-19 outcomes (Fig 4B). For example, the AUC of age combined with comorbidities in predicting patient hospitalization was 0.83/0.88 (retrospective cohort/prospective cohort) while the AUC of *hFwe-Lose* expression was 0.89/0.90 (retrospective cohort/prospective cohort) (Fig 4B). Even more strikingly, the AUC of age combined with comorbidities in predicting patient death was 0.84/0.86 (retrospective cohort/prospective cohort) while the AUC of *hFwe-Lose* expression was 0.98/0.98 (retrospective cohort/prospective cohort) (Fig 4B). Finally, we assessed the predictive value of a combinatorial approach in predicting the death of COVID-19 patients. Specifically, the AUC of all blood biomarkers (serum ferritin, CRP, D-dimer, and neutrophil-lymphocyte ratio), *hFwe-Lose* expression, age + comorbidities and the combination of blood biomarkers, age, and comorbidities was assessed. In both the retrospective and prospective cohorts, *hFwe-Lose* expression had the highest AUC (0.92) compared with blood biomarkers (0.71/0.46), age and comorbidities (0.85/0.77), or the combination of both (0.92/0.67) (Fig 4B).

## Discussion

Despite the recent availability of vaccines, in 2021, SARS-CoV-2 has continued to spread and cause numerous outbreaks worldwide. Essential to the management of this pandemic is the ability to identify and, where necessary, triage COVID-19 patients at high risk of poor disease outcomes. Unfortunately, stratifying patients based on

underlying comorbidities and/or advanced age has proved insufficient for this purpose as it fails to account for older individuals with mild disease or younger individuals who are admitted to ICU or succumb to the infection (Liu *et al*, 2020). Several blood-based markers of inflammation have been proposed as biomarkers for severe COVID-19. However, blood samples are typically taken later in the course of the disease and the prognostic values of these biomarkers have varied between studies (Cheng *et al*, 2020; Sahu *et al*, 2020). In contrast, biomarkers obtained from nasopharyngeal samples can be obtained early in disease course (i.e., when the patient first presents for SARS-CoV-2 testing) and can be obtained from a diverse array of diagnostic settings (including drive-through testing clinics).

Here, we provide the first evidence that nasopharyngeal expression of the cell fitness marker *hFwe-Lose* has significant prognostic value in COVID-19 patients. Indeed, AUC of the receiver operator characteristic curve for *hFwe-Lose* expression in predicting COVID-19 patient hospitalization and death was markedly greater than the AUC of blood biomarkers of inflammation, patient age plus comorbidities, or a combination of both. This represents the first evidence for cell fitness markers in predicting the prognosis of infectious disease.

At present, why *hFwe-Lose* expression in the respiratory tract is associated with severe COVID-19 remains unclear. It has been shown in the past that environmental changes within our body including nutrient state, inflammation, and notably, immune function impact relative fitness status of cells and their contribution to their tissue space. Chronic inflammation, including dietary-associated inflammation and obesity, antagonizes homeostatic fitness sensing mechanisms and results in higher retention of suboptimal cells, thereby worsening tissue fitness over time (Vermeulen *et al*, 2013; Sasaki *et al*, 2018; Bruens *et al*, 2020; Sato *et al*, 2020).

We have shown that in the absence of SARS-CoV-2, *hFwe-Lose* expression is elevated in the respiratory tract of older individuals and individuals with one or more underlying comorbidity. Importantly, *hFwe-Lose* expression was not elevated in *all* individuals in these patient groups and patient-to-patient differences in expression may reflect disease duration, patient lifestyles, disease management, and/or other factors. Nevertheless, we propose that on an individual level, elevated *hFwe-Lose* expression is indicative of a large number of unfit cells in the respiratory tract, and thereby typically associated with advanced age or underlying medical conditions. Specifically,

**Figure 4.** The linear regression models show superiority of the *hFwe-Lose* biomarker to predict COVID-19 outcome, compared with conventional biomarkers. ▶

A   *hFwe-Lose* predicts COVID-19 patients' death more accurately than other biomarkers. *hFwe-Lose* was compared with four known biomarkers (ferritin, CRP, D-dimer, and neutrophil-lymphocyte ratio, respectively) in predicting death of hospitalized patients. For each biomarker, only patients who had the information of the respective blood biomarker were used. 115 patients were used to compare ferritin and *hFwe-Lose*. 120 patients were used to compare D-dimer and *hFwe-Lose*. 127 patients were used to compare CRP and *hFwe-Lose*. 153 patients were used to compare neutrophil–lymphocyte ratio and *hFwe-Lose*. With AUC as the criteria, *hFwe-Lose* significantly outperformed all four biomarkers in predicting death in both retrospective and prospective cohorts. AUC coefficients, as well as CIs, are displayed on the plots.

B   *hFwe-Lose* biomarker is superior to other markers in COVID-19 poor outcome prediction. All 283 patients were used to compare *hFwe-Lose* and age combined with comorbidities in predicting hospitalization (left) and death (middle). The AUC of age combined with comorbidities in predicting the hospitalization of the prospective cohort is 0.88 (CI - [0.81–0.96]). The AUC of *hFwe-Lose* in predicting the hospitalization of the prospective cohort is 0.90 (CI - [0.84–0.97]). *hFwe-Lose* outperformed age combined with comorbidities in predicting hospitalization. The AUC of age combined with comorbidities in predicting the death of the prospective cohort is 0.86 (CI - [0.72–1.0]). The AUC of *hFwe-Lose* in predicting the death of the prospective cohort is 0.98 (CI - [0.92–1.0]). *hFwe-Lose* significantly outperformed age combined with comorbidities in predicting death. The 105 patients who registered the information of all four known blood biomarkers (ferritin, CRP, D-dimer, and neutrophil-lymphocyte ratio) were used to compare *hFwe-Lose*, age combined with comorbidities, and the four biomarkers in predicting death (right). *hFwe-Lose* significantly outperformed age combined with comorbidities and the four biomarkers in predicting death in both retrospective and prospective cohorts derived from the 105 patients.

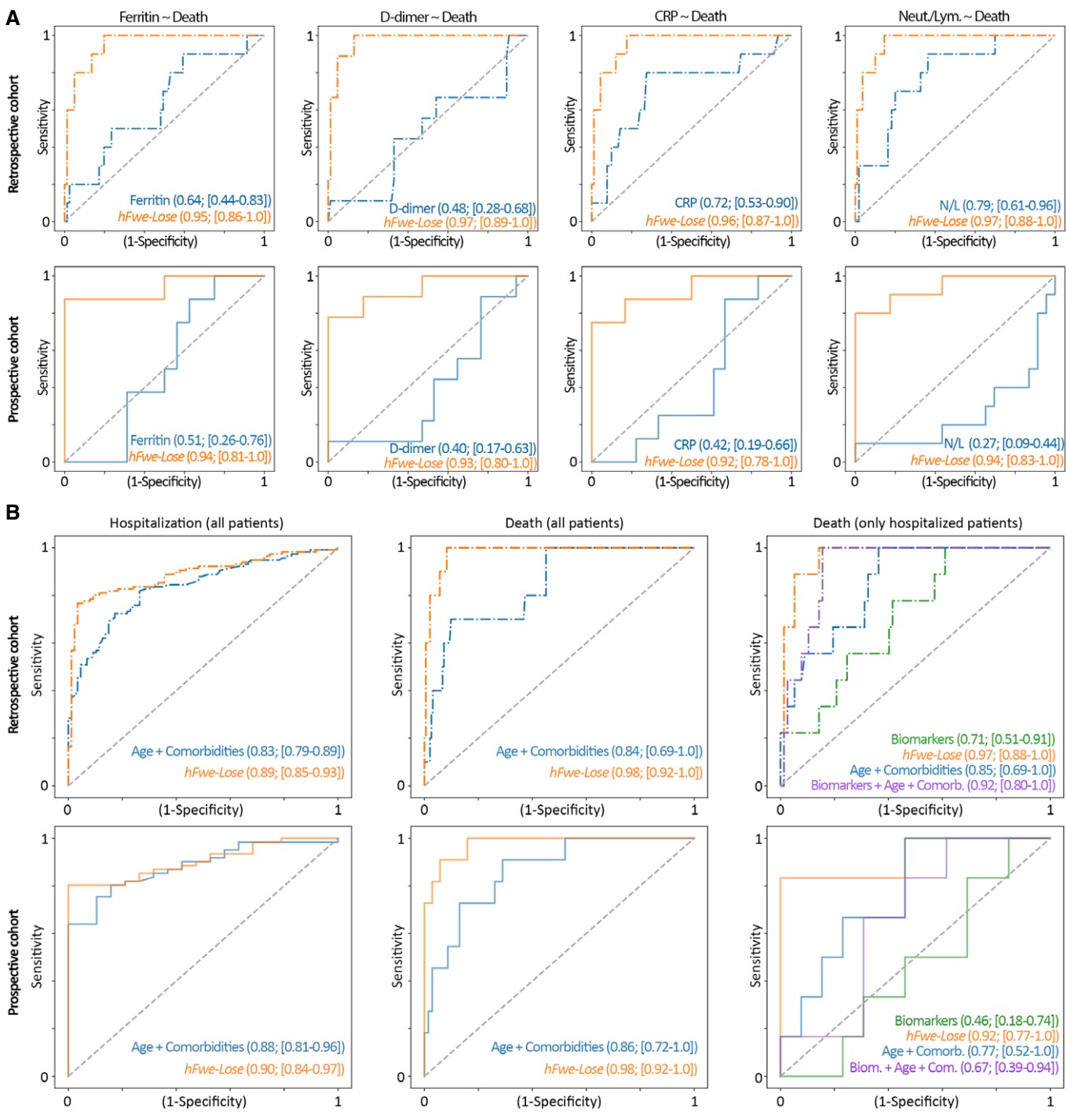

**Figure 4.**

the isoforms of the flower protein form an extracellular code that communicates fitness status between interacting cells (Rhiner *et al*, 2010; Madan *et al*, 2019). We previously found that *hFwe-Lose* is strongly induced in tumor-adjacent stromal cells that are marked for apoptotic elimination (Madan *et al*, 2019). Individuals with elevated *hFwe-Lose* expression may therefore have a larger number of cells that are susceptible to SARS-CoV-2-induced cell death. SARS-CoV-2-mediated cell death may exacerbate lung inflammation, which is a

key pathological process underpinning severe respiratory failure and death in COVID-19 patients. In this regard, *hFwe-Lose* expression may act as a "susceptibility" marker for worst outcome in those with advanced age and/or multiple comorbidities or as stand-alone risk stratification marker for those who are young and without comorbidities. For example, we predict that in young, healthy adults without underlying respiratory comorbidities, the lung alveoli tissue would typically be "fit" and would consist of very few suboptimal

cells expressing *hFwe-Lose*. Upon SARS-CoV-2 infection, due to a low number of suboptimal cells, healthy and young individuals do not suffer widespread epithelial cell death. This may result in only minor to mild symptoms and enable the patient to recover without severe complications (Appendix Fig S3). In contrast, in young adults with pre-existing suboptimal cells (for whatever reason), *hFwe-Lose* expression may be an excellent prognostic biomarker to predict outcome upon COVID-19 infection, thereby allowing clinicians to identify a subgroup of young adults who may have worst clinical outcome (as opposed to young adults who have a milder, self-limiting COVID-19 illness).

The above hypothesis remains to be tested, but it raises the intriguing possibility that *hFwe-Lose* expression could be used to screen individuals for their risk of developing severe COVID-19 *prior* to SARS-CoV-2 infection. This is not currently possible with other widely used COVID-19 biomarkers (which typically detect the host response to infection) and may serve to be a powerful tool in informing individual risk–benefit analyses of receiving a COVID-19 vaccine. There is also the possibility that, as a general marker of cell fitness, *hFwe-Lose* expression has prognostic potential for other viral pathogens. Specifically, widespread cell death and pulmonary edema is a key feature of influenza virus-induced viral pneumonia (Short *et al*, 2014). As another virus of pandemic potential, the role of cell fitness markers in the prognosis of influenza remains an area of ongoing research.

Importantly, while this study clearly demonstrated a role for *hFwe-Lose* expression in COVID-19 prognostics, there are several study limitations that are important to acknowledge. Firstly, this study was performed on a US-based population. Therefore, the prognostic use of *hFwe-Lose* expression in other geographically and ethnically diverse populations needs to be established. We also did not assess the prognostic value of *hFwe-Lose* expression in combination with other predictors of COVID-19 severity. It is possible that a combinatorial approach that encompasses *hFwe-Lose* expression would further improve the associated positive and negative predictive values.

Nevertheless, the present study established a novel prognostic biomarker for COVID-19 severity and provides the first evidence for the role of cell fitness in the pathogenesis of infectious disease.

## Materials and Methods

### RNA isolation from patient FFPE samples and qPCR

Total RNA was isolated from FFPE tissue samples using the RNeasy FFPE Kit (Qiagen). Ten nanograms of total RNA was reverse transcribed to complementary DNA (cDNA) using Superscript Vilo cDNA Synthesis Kit (Thermo Fisher) per the manufacturer's instructions. Quantitative PCR (qPCR) was performed with PowerUp SYBR Green Master Mix (Thermo Fisher) using QuantStudio 5 Real-Time PCR system. The reaction conditions included an initial denaturation step at 95°C for 2 min, followed by 40 cycles of 95°C for 15 s and 60°C for 60 s. The Ct values of samples and controls were normalized to the expression level of the GAPDH housekeeping gene. The expression of *hFwe-Lose* is given as a relative change in mRNA expression to the mean of non-hospitalized samples. All qPCRs were set up in triplicate, and the experiments were performed with at least three different

samples. The mRNA expression of h*Fwe*-Lose is relative to the mean of the hFwe-Lose mRNA expression in the disease-free lung samples. The following primers were used (F: forward; R: reverse): GAPDH: 5′-GGATGCAGGGATGATGTTC-3′ (F) and 5′-TGCACCACCAAC TGCTTAG-3′ (R); *hFwe-Lose*: 5′-GCGTGTGGATGATGATGG-3′ (F) and 5′-AGCAGAGAGTCCGTACA GCA-3′ (R).

### RNA isolation from patient nasal swabs and qPCR

Total RNA was isolated from clinical nasal swab samples using the RNeasy Micro Kit Protocol (Qiagen) following the protocol listed under "Isolation of total RNA from ejectable buccal swabs". Briefly, patient samples were mixed with kit-supplied Buffer RLT and vortexed for 1 min. The lysate was then transferred to a QIAshredder Mini Spin Column and centrifuged for 5 min at 12,000× *g* to filter debris and reduce viscosity from the samples. After adding one volume of 70% ethanol to the lysate, the sample was transferred to a RNeasy MiniElute Spin Column (Qiagen) and centrifuged for 15 s at 8,000× *g*. The lysate was washed with kit-supplied buffer RW1, DNase I-treated, and washed with 80% ethanol. Finally, the total RNA was eluted in 10 µl of RNase-free water.

RNA from patient nasal swabs was converted to cDNA and PCR-detected in one-step RT–qPCRs using the *Power*SYBR Green RNA-to-CT 1-*Step*Kit (Thermo Fisher) according to manufacturer instructions. One nanogram of RNA template was added per 10 µl reaction. The thermal cycling conditions included a reverse transcription step for 30 min at 48°C, DNA polymerase activation for 10 min at 95°C, followed by Denaturation for 15 s at 95°C, and Anneal/Extension for 1 min at 60°C for 40 cycles. All qPCRs were set up in technical triplicate. The mRNA expression of *hFwe-Lose* is relative to the mean of the *hFwe-Lose* mRNA expression in the nasopharyngeal samples from non-hospitalized patients. The following primers were used (F: forward; R: reverse): GAPDH: 5′-GGATGCAGGGATGATGTTC-3′ (F) and 5′-TGCACCACCAACTGCTTAG-3′ (R); *hFwe-Lose*: 5′-GCGTGTGG ATGATGATGG-3′ (F) and 5′-AGCAGAGAGTCCGTACA GCA-3′ (R).

### Immunohistochemistry and H&E staining

Conventional hematoxylin and eosin (H&E) staining was performed on deparaffinized sections of the FFPE autopsy asseverates of the lungs according to the current accredited staining protocol at the Institute of Medical Genetics and Pathology of the University Hospital Basel, Switzerland, as per May 2020. Immunohistochemistry was performed according to the current accredited staining protocols, applying the polyclonal ready-to-use antibody PP 229 AA from Biocare (Pacheco, CA, USA) against cleaved caspase-3 on an automated immunostainer Benchmark Ultra (Roche/Ventana, Tucson, AZ, USA), and the polyclonal antibody A0080 from Dako (Glostrup, Denmark) against fibrin(-ogen) at a dilution of 1:100,000 utilizing detection with a secondary anti-rabbit link antibody.

### Patient samples

FFPE tissue blocks of lung tissue were provided by Dr. Antonio Beltran at the Pathology Department, Champalimaud Foundation. All samples used in the study were de-identified, and FFPE archived samples with no attached patient information. Normal lung tissue and the lung tissue from COVID-19 patients were provided by Dr.

Alexandar Tzankov and were collected from deceased COVID-19 patients hospitalized at the University Hospital Basel, Switzerland, as described previously (Menter *et al*, 2020). The mean post-mortem interval from death to an autopsy was 33.3 h (11–84.5). All samples were reviewed by the institutional ethics board and determined to qualify as non-human subjects' research.

The patient samples in form of the nasopharyngeal swabs were procured from the TSB BioBank which is part of the Translational Science BioCore (TSB) affiliated with the UW Carbone Cancer Center (UWCCC), University of Wisconsin-Madison School of Medicine and Public Health, Madison, Wisconsin, USA. Informed consent was obtained from all subjects and the experiments conformed to the principles set out in the WMA Declaration of Helsinki and the Department of Health and Human Services Belmont Report. Nasopharyngeal swabs from patients positive for COVID-19 were selected based on the report from EPIC Beaker module that included results of coronavirus disease 2019 (COVID-19), PCR (UWH) test and patient demographics. Coronavirus disease 2019 (COVID-19), PCR (UWH) test was performed on nasopharyngeal swabs on Molecular Genprobe Panther Fusion platform in Molecular Diagnostics Lab. Chart review was performed on admitted patients negative for COVID-19, and patients who presented to ED with symptoms consistent with COVID-19 infection, such as fever, cough, and dyspnea, were included in the cohort. Chart review was performed on admitted patients positive for COVID-19, and patients who had only 1 positive COVID test result in their clinical history on the day of their admission were included in the cohort. The blood samples for individual patients were pulled daily for the week and then once every week for the patient duration of the hospital stay (max 1 month).

### Data analysis and biostatistics

The RT–qPCR measurements were performed in technical triplicates, while being blinded to the information about the sample. The technical triplicates (Ct values) were averaged before calculation of the dCt values. Additionally, the sample order in respective cohorts was randomized before the RT–qPCR measurements.

Data were analyzed using R 3.6.1 (R-Core-Team, 2020). *hFwe-Lose* expression, age, and outcome were analyzed using an asymptotic model of the form:

$$E(age) = A \cdot \left(1 - \exp\left(-\exp\left(r + d \cdot I_{non-hosp.}\right) \cdot age\right)\right).$$

where $E$ is the relative expression of *hFwe-Lose*, *age* is the patient's age, $A$ is the value of the asymptote, $r$ is a rate parameter, $d$ is a coefficient of the non-hospitalized group on the rate parameter, and $I_{non-hosp.}$ is an indicator variable which is 1 if the group is "non-hospitalized" and 0 otherwise. The model was fit to the retrospective data. The fit did not converge when the outcome group "death" was included, so only data from non-hospitalized and hospitalized patients were used. $R^2$ is calculated as the ratio of the residual and the total variance.

The probability of hospitalization or death and *hFwe-Lose* expression was analyzed using logistic models. Predictor variable was the logarithm of *hFwe-Lose* expression. The model included age (linear, continuous) and sex (male/female), the concentrations of blood markers (log-linear), and the presence of major known

**The paper explained**

**Problem**

Assessing the degree of risk for the development of severe COVID-19 is an important consideration in the management of the current pandemic. Such a tool would be useful for the triage of patients that test positive for COVID-19, thus enabling those likely to develop severe symptoms closer monitoring and earlier access to hospitalization and intensive care.

**Results**

We performed post-mortem analysis of COVID-19-infected lung tissues and determined that the cell fitness marker, *hFwe-Lose* can precede the host immune response to infection. More importantly, the expression levels of *hFwe-Lose* outperformed conventional methods in predicting outcomes in COVID-19 patients (hospitalization or death).

**Impact**

The cell fitness marker *hFwe-Lose* accurately predicts outcomes in COVID-19 patients. This demonstrates how tissue fitness pathways dictate the response to infection and disease, and their utility in managing the current COVID-19 pandemic.

comorbidities (yes/no). The observed statistical significance of *hFwe-Lose* expression was obtained from likelihood ratio tests of the respective coefficient within the statistical model.

The classification and regression tree (CART) was created using the "rpart" R package (Therneau & Atkinson, 2019), using standard parameters. The analysis incorporated all relevant information about patients (*hFwe-Lose* expression, age, sex, presence of comorbidities (diabetes, COPD, obesity, cardiomyopathy, heart failure, hypertension)). All patients were included in the CART analysis ($n = 283$). The number of splits was automatically determined based on the saturation of the coefficient of determination, as well as on the plateauing of the relative error.

The receiver operating characteristic (ROC) curves were created using the "ROCit" R package (Khan & Brandenburger, 2020), using standard parameters. Cut-off values were determined as the threshold value with the lowest residual between true-positive rate (TPR) and false-positive rate (FPR) curve parameters (Youden index). The cut-offs from the ROC analysis were used to create confusion matrices by comparing predicted and true outcomes.

### R packages

Attached packages are as follows: randomForest_4.6-14, party_1.3-7, strucchange_1.5-2, sandwich_3.0-1, zoo_1.8-8, modeltools_0.2-23, mvtnorm_1.1-1, rpart_4.1-15, beeswarm_0.2.3, lemon_0.4.5, ROCit_2.1.1, openxlsx_4.2.3, ggrepel_0.9.1, readxl_1.3.1, tidyr_1.1.2, plyr_1.8.6, readr_1.4.0, ggplot2_3.3.3, dplyr_1.0.4, RColorBrewer_1.1-2, viridis_0.5.1, viridisLite_0.3.0, and gplots_3.1.0.

References are available at the CRAN website "packages" section (The Comprehensive R Archive Network).

### Logistic regression with scikit-learn

Logistic regression implemented in scikit-learn (Pedregosa *et al*, 2011) 23.1 in Python 3.7.4 was used as the machine learning model

to compare *hFwe-Lose* and other factors. Class weights of all models were set as "balanced" to assign proper weight to patients. All other parameters were set with default values. L2 regularization was used by default. Following loss function was used to fit our models:

$$\min_{\omega,\, c} \frac{1}{2}\omega^T\omega + C\sum_{i=1}^{n} w_i \log\left(\exp\left(-y_i\left(X_i^T\omega + c\right)\right) + 1\right)$$

$$w_i = M_{sample}/M_{class}M_{class\, i}$$

where $\omega$ is the coefficient vector, $c$ is the intercept, and $y_i$ is the label of the $i$-th sample. The value of $y_i$ can be 1 and $-1$. $X_i$ is the input vector of the $i$-th sample. $w_i$ is the weight for the $i$-th sample. Superscript $T$ refers to transposition. The default value of $w_i$ is 1. In the "balanced" model, w of the $i$-th sample is calculated by the second equation. $M_{sample}$ is the number of all samples. $M_{class}$ is the number of class. In our case, $M_{class}$ is 2. $M_{class\, i}$ is the number of the samples that are the same class of the $i$-th sample. $C$ is a constant to specify the regularization strength. $C$ is set as 1 by default.

All 283 patients were used to compare *hFwe-Lose* and age combined with comorbidities in predicting hospitalization and death. Among the 283 patients, 203 of them are retrospective and 80 of them are prospective. Among the 203 retrospective patients, 10 of them died and 116 of them were hospitalized. Among the 80 prospective patients, 11 of them died and 61 of them were hospitalized. When comparing *hFwe-Lose*, age combined with comorbidities, and four known biomarkers in predicting death, 105 patients who had information of all the four biomarkers were used. Among the 105 patients, 86 of them are retrospective and 19 of them are prospective. Among the 86 retrospective patients, 9 of them died. Among the 19 prospective patients, 6 of them died. When comparing *hFwe-Lose* and four known biomarkers separately in predicting death, patients who had information of each biomarker were used. In the comparison between *hFwe-Lose* and Ferritin, 91 retrospective records were used for training. 10 of the 91 patients died. 24 prospective records were used for validation. 8 of the 24 patients died. In the comparison between *hFwe-Lose* and D-dimer, 94 retrospective records were used for training. 9 of the 94 patients died. 26 prospective records were used for validation. 9 of the 26 patients died. In the comparison between *hFwe-Lose* and CRP, 101 retrospective records were used for training. 10 of the 101 patients died. 26 prospective records were used for validation. 8 of the 26 patients died. In the comparison between *hFwe-Lose* and neutrophil–lymphocyte ratio, 120 retrospective records were used for training. 10 of the 120 patients died. 33 prospective records were used for validation. 10 of the 33 patients died.

# Data availability

This study includes no data deposited in external repositories.

**Expanded View** for this article is available online.

## Acknowledgements
This study was supported by Swiss Cancer League, Seeds of Science, UAMS, SNSF, Fundação para a Ciência e a Tecnologia, Fundamental Mandates (Stichting tegen Kanker—Fondation contre le Cancer) to R.G.; ERC, SNSF, Josef Steiner Cancer Research Foundation, Swiss Cancer League, and Champalimaud Foundation to E. Mo.; Novo Nordisk Foundation [NNF17CC0027852] and Lundbeck Foundation [R313–2019–421] to K.J.W.; La Caixa Funding LCF/BQ/PR20/11770006 to E. Ma.; FIS (Ministry of Health), Madrid, Spain, Grant PI17/01981 to A.L-.B.;. DFG Clinical Research Unit KFO 309, and the German Center for Lung Research (DZL), DFG Collaborative Research Center SFB1021 to J.W.; Deutsche Forschungsgemeinschaft (DFG) Clinical Research Group KFO309 TP08, the DFG-funded Excellence Cluster Cardio-Pulmonary Institute (CPI), and the German Center for Lung Research (DZL) to T.B.; Botnar Research Centre for Child Health (BRCCH) (FTC-2020-10) and SNSF to A.T., T.M., M.M. & J.D.H.; Support from the Thelma Newmeyer Corman Chair in Cancer Research, the VCU Commercialization Fund and NIH/NCI R01 CA259599 to P.B.F.; Fundação para a Ciência e a Tecnologia Grant 2020.05319.BD to A.M.P.; Fundação para a Ciência e a Tecnologia Grant SFRH/BD/139138/2018 to R.C.D.; Fundação para a Ciência e a Tecnologia and PGCD—Programa de Pós-Graduação Ciência para o Desenvolvimento *Grant* SFRH / BD/135367/2017 to D.C.; Tumor Microenvironment (TME) CoBRE Grant (NIH/NIGMS P20GM121322), West Virginia IDeA-CTR (NIH/NIGMS 2U54 GM104942-03), National Science Foundation (NSF/1920920, NSF/1761792), West Virginia IDeA Network of Biomedical Research Excellence (WV-INBRE) (NIH/NIGMS P20GM103434) to I.M.; and Adelson Medical Research Foundation, NIH P50 SPORE CA228991, Honorable Tina Brozman Foundation for Ovarian Cancer Research to R.D. The author(s) thank the Translational Science Biocore (TSB) BioBank of the University of Wisconsin Carbone Cancer Center and the clinical laboratory at the University of Wisconsin Hospitals for providing specimens and associated clinical data used in this research. The Translational Science Biocore (TSB) BioBank of the University of Wisconsin Carbone Cancer Center and the clinical laboratory at the University of Wisconsin Hospitals are supported by P30 CA014520 and received dedicated support for COVID-associated work from the University of Wisconsin School of Medicine and Public Health. We thank Taylor M. Parker for thorough reading of the manuscript. We thank Dr. Timothy Eubank at WVU for his kind support with the organization and purchase of RNA extraction kits (West Virginia Clinical and Translational Science Institute (WVCTSI) Grant GM104942). We thank Aenya Gogna for supporting the entire process involving research activity and writing of the manuscript. We thank Sunita Gogna and Keshav Chandra Gogna who suffered major COVID-19 disease, and they continuously discussed their disease progression over a period of 4 months. This helped our team in deeper understanding of the subject. We sincerely thank Mr. Luís Nunes from the Ritz Carlton, Penha Longa Resort Estrada da Lagoa Azul, Linhó, Sintra, Portugal. The Penha Longa management regularly arranged for the meeting space and facilitated important discussions between collaborators during the time of total shutdown due to the declared national emergency in Portugal.

## Author contributions
MY performed all analysis, wrote the paper, and helped with study design and concept; EM performed qPCR while remaining blind to samples and clinical information and helped with study design and concept; JW performed and supervised all biostatistical analysis, supervised and contributed to data visualization, and interpretation, wrote the manuscript, and helped in data presentation; KRS performed interpretation of the clinical and bio statistical data, wrote the manuscript, and helped with study design and concept; AMP performed qPCR while remaining blind to samples and clinical information and helped with preparation of the animations; LL performed machine learning clinical analysis, wrote the manuscript, and helped in preparation of figures; DC performed qPCR while remaining blind to samples and clinical information and helped with preparation of the animations; EN helped with procurement of sample from TSB Biobank and coordinated shipment of all samples and collection of clinical data; MTW helped with RNA extraction from

patient nasopharyngeal swab material; ESR helped with RNA extraction from patient nasopharyngeal swab material; IR helped with procurement of lung FFPE samples and collected clinical information about comorbidity status of all samples; RC-D performed qPCR while remaining blind to samples and clinical information and helped with preparation of the animations; CJP helped with manuscript writing and biostatistical analysis; MN helped with manuscript writing and biostatistical analysis; KG helped with procurement of sample from TSB Biobank and helped with organization of the manuscript and preparation of the animations; SC helped with procurement of sample from TSB Biobank and helped with organization of the manuscript and preparation of the animations; TB provided supervision to MY; RP helped with histological reading of the COVID-19 deceased patients; MSP helped with physiological readings for the CT scans of the COVID-19 deceased patients; TM collected disease-free lung patient samples and COVID-19 autopsy samples; MM collected disease-free lung patient samples and coordinated logistics and archiving; JDH collected disease-free lung patient samples and COVID-19 autopsy samples; MT helped in study design and, provided administrative support for sample collection; KDG helped with the collection of the clinical data associated with nasopharyngeal swabs, customized reports, performed chart review, and adapted clinical workflow to retrieve appropriate specimens; KAM helped with procurement of patient samples in the form of nasopharyngeal swabs from Translational Science Biocore (TSB) BioBank of the University of Wisconsin Carbone Cancer Center, SMM- helped with procurement of patient samples in the form of nasopharyngeal swabs from Translational Science Biocore (TSB) BioBank of the University of Wisconsin Carbone Cancer Center, supervised the collection of patient information; LKM helped with procurement of patient samples in the form of nasopharyngeal swabs from Translational Science Biocore (TSB) BioBank of the University of Wisconsin Carbone Cancer Center and helped with organization of the patient data collection; EAR helped with procurement of patient samples in the form of nasopharyngeal swabs from Translational Science Biocore (TSB) BioBank of the University of Wisconsin Carbone Cancer Center and helped with histological reading of COVID-19 deceased patients; ALB helped with study design and supervised; IR helped with collection of lung patient tissue samples and helped with histological reading of COVID-19 deceased patients; RD helped in organizing the procurement of RNA Extraction kits and helped with funding the research; MA helped in organizing the procurement of COVID-19 nasopharyngeal swabs and provided input on interpretation of clinical data; PBF helped in organizing the procurement of COVID-19 nasopharyngeal swabs from Translational Science Biocore (TSB) BioBank of the University of Wisconsin Carbone Cancer Center and helped with funding the research; SRG helped in organizing the procurement of COVID-19 nasopharyngeal swabs from Translational Science Biocore (TSB) BioBank of the University of Wisconsin Carbone Cancer Center and helped with funding the research; AKG helped with manuscript writing supported study design and provided key input on interpretation of clinical data; AK helped with interpretation of clinical and bio statistical data and helped in writing the manuscript; IM helped with RNA extraction from patient nasopharyngeal swab material and supervised MTW and ESR; CBM helped with manuscript writing supported study design and provided key input on interpretation of clinical data; BT helped interpretation of clinical and biostatistical data and helped in writing the manuscript; MSW helped with manuscript writing supported study design and provided key input on interpretation of clinical data; KJW supervised LL, supervised and designed machine meaning tools to analyze clinical results, funded the research, and provided organizational support; AT helped in study design, provided administrative support for sample collection, collected disease-free lung patient samples and COVID-19 autopsy samples, performed histopathologic analysis and caspase IHC, partially wrote the manuscript, funded the research, provided organizational support, and helped with procurement of COVID-19 nasopharyngeal swabs from Translational Science Biocore (TSB) BioBank of the University of Wisconsin Carbone Cancer Center; EM provided the initial ideas for the project, helped with study design and manuscript writing, funded the research, and provided organizational support; RG designed the study, provided the initial ideas for the project, designed the study concept, funded the research, wrote the manuscript, procured patient samples, supervised the research, and established all required international collaborations.

## Conflict of interest

The authors declare that they have no conflict of interest.

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

## List of affiliations

Michail Yekelchyk[1]; Esha Madan[2]; Jochen Wilhelm[3,4]; Kirsty R Short[5]; António M Palma[2]; Linbu Liao[6]; Denise Camacho[2]; Everlyne Nkadori[7]; Michael T Winters[8]; Emily S Rice[8]; Inês Rolim[2]; Raquel Cruz-Duarte[9]; Christopher J Pelham[10]; Masaki Nagane[11]; Kartik Gupta[12]; Sahil Chaudhary[12]; Thomas Braun[1,13]; Raghavendra Pillappa[14]; Mark S Parker[15]; Thomas Menter[16]; Matthias Matter[16]; Jasmin Dionne Haslbauer[16]; Markus Tolnay[16]; Kornelia D Galior[7]; Kristina A Matkwoskyj[7]; Stephanie M McGregor[7]; Laura K Muller[7]; Emad A Rakha[17]; Antonio Lopez-Beltran[2,18]; Ronny Drapkin[19,20,21]; Maximilian Ackermann[22,23]; Paul B Fisher[24,25,26]; Steven R Grossman[27,28]; Andrew K Godwin[29,30]; Arutha Kulasinghe[31]; Ivan Martinez[8]; Clay B Marsh[8]; Benjamin Tang[32]; Max S Wicha[33,34]; Kyoung Jae Won[6,35]; Alexandar Tzankov[16]; Eduardo Moreno[2]; Rajan Gogna[2,6,35]

[1]Department of Cardiac Development and Remodelling, Max Planck Institute for Heart and Lung Research, Bad Nauheim, Germany. [2]Champalimaud Centre for the Unknown, Lisbon, Portugal. [3]Universities Giessen & Marburg Lung Center, German Center for Lung Research (DZL), Justus-Liebig-University, Giessen, Germany. [4]Institute for Lung Health (ILH), Universities Giessen & Marburg Lung Center, German Center for Lung Research (DZL), Justus-Liebig-University Giessen, Giessen, Germany. [5]School of Chemistry and Molecular Biosciences, The University of Queensland, Brisbane, Qld, Australia. [6]Biotech Research and Innovation Centre (BRIC), University of Copenhagen, Copenhagen N, Denmark. [7]Department of Pathology and Laboratory Medicine, University of Wisconsin Carbone Cancer Center, University of Wisconsin-Madison School of Medicine and Public Health, Madison, WI, USA. [8]Department of Microbiology, Immunology & Cell Biology and WVU Cancer Institute, West Virginia University, Morgantown, WV, USA. [9]Instituto de Medicina Molecular João Lobo Antunes, Faculdade de Medicina, Universidade de Lisboa, Lisboa, Portugal. [10]Eurofins Panlabs Inc., St. Charles, MO, USA. [11]Department of Biochemistry, School of Veterinary Medicine, Azabu University, Kanagawa, Japan. [12]Department of Surgery, School of Medicine and Public Health, University of Wisconsin, Madison, WI, USA. [13]Member of the German Center for Cardiovascular Research (DZHK), Greifswald, Germany. [14]Department of Pathology, Virginia Commonwealth University School of Medicine, Richmond, VA, USA. [15]Department of Diagnostic Radiology and Internal Medicine, Early Detection Lung Cancer Screening Program, Thoracic Imaging Division, Thoracic Imaging Fellowship Program, VCU Health Systems, Richmond, VA, USA. [16]Pathology, Institute of Medical Genetics and Pathology, University Hospital Basel and University of Basel, Basel, Switzerland. [17]Division of Cancer and Stem Cells, Department of Pathology, School of Medicine, Nottingham University Hospitals, University of Nottingham, Nottingham, UK. [18]Department of Morphological Sciences, Cordoba University, Cordoba, Spain. [19]Penn Ovarian Cancer Research Center, Department of Obstetrics and Gynecology, University of Pennsylvania Perelman School of Medicine, Philadelphia, PA, USA. [20]Graduate Program in Cell and Molecular Biology, University of Pennsylvania Perelman School of Medicine, Philadelphia, PA, USA. [21]Basser Center for BRCA, Abramson Cancer Center, University of Pennsylvania School of Medicine, Philadelphia, PA, USA. [22]Institute of Pathology and Molecular Pathology, Helios University Clinic Wuppertal, University of Witten/Herdecke, Wuppertal, Germany. [23]Institute of Functional and Clinical Anatomy, University Medical Center of the Johannes Gutenberg-University Mainz, Mainz, Germany. [24]Department of Human and Molecular Genetics, School of Medicine, Virginia Commonwealth University, Richmond, VA, USA. [25]Massey Cancer Center, Virginia Commonwealth University, Richmond, VA, USA. [26]Department of Human and Molecular Genetics, Institute of Molecular Medicine, School of Medicine, Virginia Commonwealth University, Richmond, VA, USA. [27]Department of Internal Medicine, Keck School of Medicine, Norris Comprehensive Cancer Center, Los Angeles, CA, USA. [28]University of Southern California, Los Angeles, CA, USA. [29]Department of Pathology and Laboratory Medicine, University of Kansas Medical Center, Kansas City, KS, USA. [30]University of Kansas Cancer Center, Kansas City, KS, USA. [31]The University of Queensland Diamantina Institute, The University of Queensland, Brisbane, Qld, Australia. [32]Department of Intensive Care Medicine, Nepean Hospital, Penrith, NSW, Australia. [33]Rogel Cancer Center, University of Michigan, Ann Arbor, MI, USA. [34]Department of Internal Medicine, Michigan Medicine, University of Michigan, Ann Arbor, MI, USA. [35]Faculty of Health and Medical Sciences, Novo Nordisk Foundation Center for Stem Cell Biology, DanStem, University of Copenhagen, Copenhagen N, Denmark.

