## [Review Process File · EMBO Molecular Medicine]

Flower Lose, a Cell Fitness Marker, Predicts COVID-19 Prognosis

Michail Yekelchik, Esha Madan, Jochen Wilhelm, Kirsty Short, António Palma, Linbu Liao, Denise Camacho, Everlyne Nkikadori, Michael Winters, Emily Westemeier, Inês Rolim, Raquel Cruz-Duarte, Christopher Pelham, Masaki Nagane, Kartik Gupta, Sahil Chaudhary, Thomas Braun, Raghavendra Pillappa, Mark Parker, Thomas Menter, Matthias Matter, Jasmin Haslbauer, Markus Tolnay, Kornelia Galior, Kristina Matkwoškyj, Stephanie McGregor, Laura Muller, Emad Rakha, Antonio Beltran, Ronny Drapkin, Maximilian Ackermann, Paul Fisher, Steven Grossman, Andrew Godwin, Arutha Kulasinghe, Ivan Martinez, Clay Marsh, Benjamin Tang, Max Wicha, Kyoung Won, Alexandar Tzankov, Eduardo Moreno, and Rajan Gogna

DOI: 10.15252/emmm.202013714

Corresponding author(s): Rajan Gogna (rajangogna@gmail.com), Alexandar Tzankov (alexandar.tzankov@usb.ch), Eduardo Moreno (eduardo.moreno@research.fchampalimaud.org)

Review Timeline:

Submission Date:	10th Nov 20
Editorial Decision:	17th Nov 20
Revision Received:	2nd Aug 21
Editorial Decision:	11th Aug 21
Revision Received:	14th Sep 21
Accepted:	16th Sep 21

Editor: Zeljko Durdevic

Transaction Report:

Dear Rajan,

Thank you for your interest and submitting your manuscript (EMBOJ-2020-106690) for consideration by the EMBO Journal. Please again accept my apologies for the unusual delay due to protracted referee input. Your study has been sent to four referees for evaluation and we have received reports from all of them, which I enclose below. Please note that the reviewers cover complementary areas of expertise: Referees #1 and #3 are experts in cell competition, referee #2 a broader infection biologist, and referee #4 an airway translational biomarker researcher. In light of their comments, I am afraid we decided that we cannot offer publication in The EMBO Journal.

As you can see, the referees appreciate the matter addressed and that the analysis extends previous work. However they at the same time raise major concerns that I am afraid preclude publication here. In particular, referee #4 state major caveats regarding the prospective clinical value and prognostic meaning of your results, which we agree is a key limitation. We have discussed the reports in depth. In light of all information at hand, I am afraid we have concluded that we cannot offer to publish your study in The EMBO Journal. I still hope that you will find the referees' comments helpful.

I regret that I cannot be more positive in this case. I thank you again for your sharing the work with us.

Kind regards,

Daniel

Daniel Klimmeck, PhD
Editor
The EMBO Journal

Referee #1:

This manuscript attempts to persuade the reader that, in their words, "expression of Flower Lose (hFweLose), a unique cell-fitness biomarker, is accumulated in older adults and adults with comorbidities." And, that "hFweLose expression provides an assessment of the fitness of cells in the lung tissue of an individual, and it can accurately predict the worst outcome for COVID-19 patients." The authors use a variety of correlative tests of hFwe mRNA expression from published RNA seq data (GTEx of CACFD1, the official name of hFwe; it is not clear which hFwe isoform is measured), and autopsy samples from a small cohort of COVID patients with or without known comorbidities. They also carry out laser capture of stromal, tumor and normal cells from a small number of autopsy samples followed by RT-PCR to look at hFweLose mRNA expression. Many correlative examples of expression of hFwe and hFweLose mRNA are used to support their argument. Some of the correlations are

interesting. For example, that hFweLose expression seems to correlate with age in Fig. 1M. In Fig. 1N, the correlation of hFweLose expression vs COVID 19 vs comorbidities is also interesting. The authors use each correlation to then make additional correlations by a variety of different methods to build their case, culminating with a PCA. To the authors, the PCA serves to provide evidence that hFweLose expression could be a biomarker that predicts COVID morbidity. However, it is important to point out that PCA is useful for exploring data - describing it - but is not appropriate to use as "evidence".

Indeed, is the correlation between hFweLose expression and COVID /susceptibilities and deaths even valuable? One could argue that many genes, picked at random from the genome, could show this kind of correlation. Did the authors carry out this kind of meta analysis on any other genes? Importantly, how do the authors envision that their model works mechanistically? They claim, on p. 5, "... it would therefore be more feasible to quarantine people with high expression of hFweLose, rather than all individuals above the age of 70, in addition to individuals with comorbidities, which may constitute a large percentage of the world population." How do they propose that hFweLose should be a biomarker for COVID susceptibility? Would they advocate performing lung biopsies on everyone to see how much hFweLose they express? This is not practical; it is also not clear how this would improve predictions made from the presence of comorbidities, for example. Overall the paper does not convince me that hFweLose is a good marker for COVID. Numerous additional reasons are addressed below.

Are the conclusions of the paper justified based on the presented data? In general, my answer to this question is no. Correlative data does not indicate causation, and important tests are missing from their analysis. First, the sample size of 21 autopsies, 11 of which were used here, is extremely small (despite the claim on p. 4, that "...we performed the largest autopsy study on the COVID-19 death cases. In this study, we included the lung autopsy samples from 11 patients, who were diagnosed with and died from COVID-19 (Menter et al., 2020)."). For the conclusions to be meaningful thousands of samples would be needed. In addition, many comparisons are made between an even smaller cohort of disease-free patient samples that are not age-matched with the diseased patient samples, nor accounted for in terms of pre-dispositions, environmental factors, etc. The data do not appear to be analyzed by a statistician who understands what can be actually be appropriately compared.

Throughout the paper, the authors tend to make very strong statements that are not based in fact, are clearly overstated, and/or sometimes hyperbolic. I highlight this problem with some examples:

1. "We have identified that the expression of a unique cell fitness biomarker, hFweLose, underlies important comorbidities such as diabetes, obesity, chronic obstructive pulmonary disease (COPD), liver cirrhosis and ageing". No references are provided for this statement, and no data is provided that expression of hFweLose actually underlies (i.e., is causal).
2. "In COVID-19 patients, suboptimal cells with low fitness status in the lung tissue are at high risk of SARS-CoV-2-mediated apoptosis" No references are provided for this statement.
3. "Currently, the only known direct cell-cell fitness sensing mechanism is via Flower, a transmembrane protein, recognized as a fitness mark, or fitness fingerprint (Rhiner et al., 2010)." This statement is incorrect: a) The mechanism of Fwe function is completely unknown; b) several other markers of cell-cell fitness have been identified (e.g., redox factors, pro-apoptotic factors, NFkB factor activity, TNF activity, etc); and c) in the ten years since publication of the Rhiner paper, there does not appear to be any studies of the "Fwe code" in cell fitness/cell competition by any group outside of the senior authors. The lack of this test of independent confirmation makes it difficult to assess its importance.
4. "We present evidence in support of our hypothesis that age and comorbidities have a massive impact on lung tissue fitness and resistance to SARS-CoV-2 assisted apoptosis." As

far as I am aware this hypothesis is widespread among COVID experts.

5. These results suggest that the upregulation of hFwe is caused by the upregulation of hFweLose, which compromises lung tissue fitness and function upon SARS-CoV-2 infection." I do not understand how this conclusion (my underlining) can possibly be made based on the data shown.

6. Conclusions of data that are presented out of context and many lack appropriate controls. As an example: "To measure the degree of apoptosis of respiratory cells, we stained for active/cleaved caspase-3, an apoptosis marker, in SARS-CoV-2-infected lung tissue. We found that 11.7% of cells, particularly in the interstitium, displayed high caspase-3 positivity and another 19.8% were low; 31.5% caspase positive cells in total (Fig 2H, left and right). The caspase staining was present in the form of patches, with the presence of both caspase-positive and caspase-negative regions. We laser captured such regions and observed the expression of hFweLose and found that the caspase-positive regions had near 6-fold high expression than the caspase-negative regions". There are no controls, e.g., in non-diseased tissues, shown for these data, making the data somewhat meaningless. Another example: "In all three analyzed tissues, hFwe gene expression was significantly upregulated in the tumor-adjacent stroma compared to normal tissue (Fig 1C)." The disease-free samples were mostly from younger people whereas that from patients with comorbidities and COVID19 were older. Does this suggest that hFweLose exp increases with age, irrespective of disease?

7. Confusion regarding the authors model of how Fwe functions as a marker of fitness in cell-cell interactions. "...expression of the fitness marker hFwe was upregulated in lung tissue when compared with other examined tissues and that aged lung tissue had an upregulation of hFweLose, which correlates with a higher number of suboptimal cells." There are several problems with this and similar statements regarding the expression of hFwe and hFweLose in this study. As the senior authors have claimed in previous studies, one cannot interpret meaning from the expression or effect of hFweLose out of context - it is necessary to know the Fwe isoform status is in nearby cells. for example, from the Madan et al 2019, which is cited often in this work, "cells expressing hFWE1 or hFWE3 undergo cell death only when co-cultured with cells expressing either hFWE2 or hFWE4". The authors have made a similar point in their publications using *Drosophila*: hFweLose only results in cell autonomous cell death when in presence of the other Fwe isoforms. Without quantifying the level of 'non-lose' isoforms of hFwe, it is impossible to know if hFweLose expression in the different tissues examined here is significant or of any consequence. In addition, the model presented by the authors on pages 6-7 does not make sense. According to their published model of Flower (Madan 2019, Rhiner 2010, etc), hFweLose doesn't need a SarsCoV2 infection to kill cells - in the right context, hFweLose is supposed to do that on its own.

8. Some arguments made by the authors are confusingly circular. For example, they state "Yet, although comorbidities are associated with increased case severity, patient risk is confounded by many variables. Therefore, the presence of co-existing disease and age alone cannot predict viral disease severity." The logic here seems to be flawed: they argue that hFweLose is a good fitness marker for Covid-19 because its expression is associated with COVID-19 comorbidities, while at the same time arguing that such comorbidities are not enough to predict disease severity.

Minor concerns that should be addressed:

1. Were the disease-free controls also from autopsies? This is not made clear. Also not clear but relevant is the cause of the disease-free patient deaths and any compromising factors (e.g., non-smokers vs smokers, lung trauma, etc) they might have had.

2. The legend to Fig. 1D says that tissues are "matched". The implication here is that each stromal area is matched to the tumor it surrounds it. How can this stroma be matched to normal lung (where there is no tumor)?

3. Fig. 2F legend does not match the figure.
4. What were the contributions of each author?
5. Many references are incomplete (no journal, date, etc).
6. The methods are very sparsely written, making it difficult to understand how many of the analyses were actually done.

Referee #2:

This manuscript by Yekelchik et al. examines the value of hFweLose as a fitness mark in COVID -19. This study is from a group that has been studying Flower proteins in the context of cell fitness and apoptosis for some time; first in *Drosophila* and later in human cancer. The overall hypothesis that susceptible individuals with predisposing conditions have more "unfit" cells is reasonably interesting even if very descriptive.

This manuscript would be greatly improved by removing about 40% of the content, which is not original or meaningful and explaining things in more detail especially in the legends.

1. Figure 1 A is quite obviously a schematic, but please state that in the legend.
2. Why the emphasis on tumors in Figure 1? Why is this relevant in a manuscript about COVID-19? This topic was covered well in the Madan et al. paper in Nature in 2019 from the same group. No really relevant new information here, and the data here would be more appropriate for another cancer related paper.
3. Figure 1B is totally informative, in no way reflects "cross-talk" as labeled and should be deleted
4. Figure 1E is a schematic which adds no information to concepts that could be easily gleaned from the data in Figure 1D. This issue has been covered before and does not belong in this manuscript.
5. The information in Figure 1F could be easily just covered by referring to the literature and does not belong in a primary paper,
6. The information in the figure legends needs to be made more clear and with explicit information as to which data bases information was obtained for each panel from 1G to 1N
7. The statement about the "largest" autopsy series with just 11 patients is odd. The legend to figure 2A suggests that 249 COVID patients were studied. What is the real information here? There needs to be far more information right through- everything is very cryptic and poorly explained
8. Including the CT scans from 2 patients is totally unnecessary. Plenty of published CT scans already. Fig 2F should be deleted
9. Showing the apoptosis in 2G could be considered relevant - bit please show high power views.

Referee #3:

The study by Michail Yekelchik, Esha Madan and colleagues investigates a possible

correlation between cell competition failure and Covid-19 severity. In particular, they analyse the expression of a well-characterised marker of cell competition, Flower (Fwe), whose Lose isoform (hFweLose) is known to tag suboptimal cells for apoptotic death in human cancers. Apoptosis is a major challenge in MERS and SARS, and the consistent tissue damage results in severe adverse symptoms. The main finding of this study is that hFweLose expression increases with ageing, hypertension, diabetes, obesity and other conditions considered comorbidity factors in Covid-19 patients. The presence of one or more of these conditions is known to exacerbate the disease, and this correlates with a consistent increase of hFweLose expression, setting it as a bona-fide predictive marker for disease outcome.

The study is of high interest as it unveils a novel, relevant role for cell competition (more generally, for homeostatic tissue regulation) in the response to infectious disease. The authors hypothesise that healthy lungs (such as other organs) be capable to promptly identify and eliminate unfit cells, so maintaining organ fitness at the best, but the presence of too many suboptimal cells in ageing or sick lungs disrupts this process, and allows accumulation of unfit cells, making lungs susceptible to a worse disease progression. Although the reason why elders, along with fragile patients, respond poorly to SARS-CoV-2 aggression seems obvious at the intuitive level, findings are missing about specific mechanisms and molecules which can underlie individual vulnerability. The authors are a distinguished voice in the topic of cell competition and their premises are convincing. Methods are briefly but adequately described, results are clear, statistical analysis is appropriate and the overall conceptual and experimental scheme is robust.

The manuscript necessitates some minor amendments before being accepted for publication, as follows:

Results, page 3:

Please insert: (Fig. 1E) after the sentence: This elimination of stromal cells creates space and availability of nutrients for cancer to grow and dominate the tissue space.

Results, page 5:

In the second line the authors refer to 21 patients and cite the same study as earlier in the text, where they refer to 11 patients. Please conform patient number between the two statements.

Fig. 1F

ICU=Intensive Care Unit, please specify it in the legend.

Fig. 2F

Numbers and letters in the figure legend do not match with those reported in the figure. Please fix it.

Fig. 2G

Readers may not be confident with lung histology, so please include a detailed frame of a normal lung specimen. In G1, i = Cell elimination: please explain how you recognise the encircled cells are being eliminated. In G2, does "The adjacent interstitial space shows analogous changes to 2B1" refer to Figure "2F"? See above.

Fig. 2H: please indicate with an arrow or similar the two apoptotic cells in the upperleft alveolum.

The legend for FweLose expression analysis in Cas-pos and Cas-neg cells is completely missing.

Summary:

In the concluding sentence, the authors claim FweLose expression may be readily detected in patient samples, but how do they think to assess it in Covid-19 patients? What tissues/cells should be processed to get reliable risk predictions? The authors should discuss this, or mitigate this affirmation.

Some typos are found across the text, please go carefully through it and fix them.

Referee #4:

The authors have done an extremely rigorous evaluation of flower lose expression in lung tissue as a predictor of Covid -19 severity and prognosis. Unfortunately they have failed to perform the one clinical correlation that has prognostic meaning e.g. measuring flower lose expression in peripheral blood and correlating that with expression in the lung and clinical outcome in an actual prospective study. Autopsy studies and cellular studies are of some value but what every clinician wants to know is whether the Flower Lose expression adds to clinical predictors of outcome and hence can be used to justify early and aggressive treatment. Based on this standard, calling this a clinical marker of prognosis is over reaching.

major comment #1 what you have done is shown that Flower Lose expression correlates with clinical events in vitro but you have no evidence of its predictive value in vivo in living subjects. I think you need to tone down the predictive claims here since there is absolutely no evidence of prediction in a clinical scenario in a living subject.

Major comment #2: What is the relationship between Flower Lose expression in lung tissue and Flower Lose expression in the serum or plasma of a living patient with Covid-19 infection?

Major comment #3: Does Flower Lose expression in peripheral blood in a large cohort of Covid-19 infected patients have any predictive power with regard to clinical outcomes?

Major Comment #4: If the answer to comment #3 is yes is it independent of comorbidities e.g. age, BMI, sex, etc.

These are the questions you need to answer. What you have shown is that there is an in vitro correlation of Flower Lose expression in the lung in patients who have died from Covid 19.

17th Nov 2020

Dear Dr. Gogna,

Thank you for the submission of your manuscript to EMBO Molecular Medicine. The referees acknowledge the interest of the study but also raise some concerns that should be addressed in a major revision of the current manuscript. As discussed with you, the focus of the revision should be on strengthening the translational/clinical aspect of the study. We agreed that cancer related data should be removed and that the revision should include both retrospective and prospective clinical study with a large patient nasal swab sample population as outlined in your revision plan.

Addressing the reviewers' concerns in full, experimentally or in writing, will be necessary for further considering the manuscript in our journal, and acceptance of the manuscript will entail a second round of review. EMBO Molecular Medicine encourages a single round of revision only and therefore, acceptance or rejection of the manuscript will depend on the completeness of your responses included in the next, final version of the manuscript. For this reason, and to save you from any frustrations in the end, I would strongly advise against returning an incomplete revision.

I look forward to receiving your revised manuscript.

Yours sincerely,

Zeljko Durdevic

Point-by-point response to the reviewers' comments

Referee #1:

1. This manuscript attempts to persuade the reader that, in their words, "expression of Flower Lose (hFweLose), a unique cell-fitness biomarker, is accumulated in older adults and adults with comorbidities." And, that "hFweLose expression provides an assessment of the fitness of cells in the lung tissue of an individual, and it can accurately predict the worst outcome for COVID-19 patients." The authors use a variety of correlative tests of hFwe mRNA expression from published RNA seq data (GTEx of CACFD1, the official name of hFwe; it is not clear which hFwe isoform is measured), and autopsy samples from a small cohort of COVID patients with or without known comorbidities.

Response

As suggested by this reviewer and others we have removed the data dealing with the correlative tests of hFwe mRNA expression from published RNA-seq data GTEx of CACFD1. The expression of *hFwe-Lose* is observed in the autopsy samples of the COVID patients and the information about their comorbidities is provided in Table S1.

They also carry out laser capture of stromal, tumor and normal cells from a small number of autopsy samples followed by RT-PCR to look at hFweLose mRNA expression.

Response

As suggested by this reviewer and others we have removed the data related to the expression of *hFwe-Lose* mRNA in stromal, tumor and normal cells. A large retrospective and prospective trial is added to the study where expression of *hFwe-Lose* is observed in the epithelial cells from the buccopharyngeal region present in the nasal swabs of the COVID-19 patients.

2. Many correlative examples of expression of hFwe and hFweLose mRNA are used to support their argument. Some of the correlations are interesting. For example, that hFweLose expression seems to correlate with age in Fig. 1M. In Fig. 1N, the correlation of hFweLose expression vs COVID 19 vs comorbidities is also interesting.

Response

We thank the reviewer for finding these correlations interesting and we have added more robust data towards these findings.

3. The authors use each correlation to then make additional correlations by a variety of different methods to build their case, culminating with a PCA. To the authors, the PCA serves to provide evidence that hFweLose expression could be a biomarker that predicts COVID morbidity. However, it is important to point out that PCA is useful for exploring data - describing it - but is not appropriate to use as "evidence".

Response

We thank the reviewer for this very important suggestion. In the revised manuscript we have performed direct experiments with a large number of patient samples in both retrospective and the prospective setting. We have removed old PCA analysis as the only basis to support our findings.

4. Indeed, is the correlation between hFweLose expression and COVID /susceptibilities and deaths even valuable? One could argue that many genes, picked at random from the genome, could show this kind of correlation. Did the authors carry out this kind of meta-analysis on any other genes?

Response

We think that the point raised by the reviewer is very valid. Although we have removed this section of the data from the revised manuscript, we wanted to check this point raised by the respected reviewer, and hence we have prepared some additional analysis, which is presented below, but is not part of the revised manuscript.

We have used published lists of differentially expressed genes upon Asthma (Airway smooth muscle cells, GSE63744), Cardiovascular disease (Myocardium, GSE116250), COPD (Alveolar macrophages, Bronchial epithelium, Peripheral blood, GSE12418) and Ageing (Gingival tissue, GSE83382). These are examples of the common comorbidities of the COVID-19, which have the expression profiles in respective tissues published. All lists were filtered to only include the DEGs, which were statistically significant (FDR < 0.05). To investigate whether there are some other genes, which share the differential expression pattern with the *hFwe-Lose*, we compared all DEGs lists and found no intersection (Rev. Fig. 1). CVD and Ageing had the highest overlap (3.3%), then were Ageing and Asthma (0.9%) and CVD and Asthma (0.8%). Altogether, it suggests that there are no universal genes, which would share the same pattern across many comorbidities and ageing, like *hFwe-Lose*.

Rev. Fig. 1. Comparison of the DEGs between various publicly available datasets of the common COVID-19 comorbidities.

5. Importantly, how do the authors envision that their model works mechanistically? They claim, on p. 5, "... it would therefore be more feasible to quarantine people with high expression of hFweLose, rather than all individuals above the age of 70, in addition to individuals with comorbidities, which may constitute a large percentage of the world population." How do they propose that hFweLose should be a biomarker for COVID susceptibility? Would they advocate performing lung biopsies on everyone to see how much hFweLose they express? This is not practical; it is also not clear how this would improve predictions made from the presence of comorbidities, for example. Overall the paper does not convince me that hFweLose is a good marker for COVID. Numerous additional reasons are addressed below.

Are the conclusions of the paper justified based on the presented data? In general, my answer to this question is no. Correlative data does not indicate causation, and important tests are missing from their analysis. First, the sample size of 21 autopsies, 11 of which were used here, is extremely small (despite the claim on p. 4, that "...we performed the largest autopsy study on the COVID-19 death cases. In this study, we included the lung autopsy samples from 11 patients, who were diagnosed with and died from COVID-19 (Menter et al., 2020)."). For the conclusions to be meaningful thousands of samples would be needed. In addition, many comparisons are made between an even smaller cohort of disease-free patient samples that are not age-matched with the diseased patient samples, nor accounted for in terms of pre-dispositions, environmental factors, etc. The data do not appear to be analyzed by a statistician who understands what can actually be appropriately compared.

Response

We agree with the reviewer. The old version of the manuscript is revised on the lines suggested by this reviewer. In the revised version we have made great effort to foster a network of international collaborations and recruit a large study of 283 patients in the retrospective and prospective setting. In this study we have observed the expression of *hFwe-Lose* in the epithelial cells of the nasal swabs from these individuals. As suggested by the reviewer the data in the revised manuscript is analyzed with help of an expert biostatistician, Dr. Jochen Wilhelm. Other biostatisticians such as Dr. Benjamin Tang also analyzed and contributed to the revised version of the MS. In addition, the manuscript is analyzed by top machine learning artificial intelligence scientist Dr. KJ Won.

6. Throughout the paper, the authors tend to make very strong statements that are not based in fact, are clearly overstated, and/or sometimes hyperbolic. I highlight this problem with some examples:

a. "We have identified that the expression of a unique cell fitness biomarker, *hFweLose*, underlies important comorbidities such as diabetes, obesity, chronic obstructive pulmonary disease (COPD), liver cirrhosis and ageing". No references are provided for this statement, and no data is provided that the expression of *hFweLose* actually underlies (i.e., is causal).

Response

We agree with the reviewer. We have removed these statements.

b. "In COVID-19 patients, suboptimal cells with low fitness status in the lung tissue are at high risk of SARS-CoV-2-mediated apoptosis" No references are provided for this statement.

Response

We agree with the reviewer. We have removed these statements.

c. "Currently, the only known direct cell-cell fitness sensing mechanism is via Flower, a transmembrane protein, recognized as a fitness mark, or fitness fingerprint (Rhiner et al., 2010)." This statement is incorrect: a) The mechanism of Fwe function is completely unknown; b) several other markers of cell-cell fitness have been identified (e.g., redox factors, pro-apoptotic factors, NFkB factor activity, TNF activity, etc); and c) in the ten years since publication of the Rhiner paper, there does not appear to be any studies of the "Fwe code" in cell fitness/cell competition by any group outside of the senior authors. The lack of this test of independent confirmation makes it difficult to assess its importance.

Response

hFwe-Lose is a leading cell fitness biomarker as its expression as a transmembrane protein accurately predicts cell survival via direct cell to cell contact (Rhiner 2010, Madan 2019). Redox signaling, NF-kB activity and TNF activity have been implicated in cell competition mechanisms to varying degrees, and to our knowledge, have not been demonstrated to function as *direct*

markers of cellular fitness status. Secondly, to our knowledge, these pathways have been studied specifically in *Drosophila* and their role in human tissue is lacking. Mechanisms such as redox signaling, NF- κ B, apoptosis and TNF may communicate cell fitness status, and their overlap with other unrelated processes such as inflammation make them poor candidates for biomarkers.

d. "We present evidence in support of our hypothesis that age and comorbidities have a massive impact on lung tissue fitness and resistance to SARS-CoV-2 assisted apoptosis." As far as I am aware this hypothesis is widespread among COVID experts.

e. These results suggest that the upregulation of hFwe is caused by the upregulation of hFweLose, which compromises lung tissue fitness and function upon SARS-CoV-2 infection." I do not understand how this conclusion (my underlining) can possibly be made based on the data shown.

Response

We agree with the reviewer. We have removed these statements.

f. Conclusions of data that are presented out of context and many lack appropriate controls. As an example: "To measure the degree of apoptosis of respiratory cells, we stained for active/cleaved caspase-3, an apoptosis marker, in SARS-CoV-2-infected lung tissue. We found that 11.7% of cells, particularly in the interstitium, displayed high caspase-3 positivity and another 19.8% were low; 31.5% caspase positive cells in total (Fig 2H, left and right). The caspase staining was present in the form of patches, with the presence of both caspase-positive and caspase-negative regions. We laser captured such regions and observed the expression of hFweLose and found that the caspase-positive regions had near 6-fold high expression than the caspase-negative regions". There are no controls, e.g., in non-diseased tissues, shown for these data, making the data somewhat meaningless.

Response

We have added appropriate controls as suggested by the reviewer. We have performed Caspase-3 staining in normal lung tissue from non-diseased individuals.

g. Another example: "In all three analyzed tissues, hFwe gene expression was significantly upregulated in the tumor-adjacent stroma compared to normal tissue (Fig 1C)."

Response

As suggested by this reviewer and others this data is removed from the revised MS.

h. The disease-free samples were mostly from younger people whereas that from patients with comorbidities and COVID19 were older. Does this suggest that hFweLose exp increases with age, irrespective of disease?

Response

In the revised manuscript we have included a large number of samples (n = 96) from disease-free. The point raised by the reviewer is very important. It does appear that *hFwe-Lose* expression increases in the lung tissue with age, although age and comorbidity are not independent variables. It means that older people tend to have more comorbidities. However, we have prepared an analysis for the reviewer where we have only included samples with no comorbidities or only one comorbidity and identified the expression value of *hFwe-Lose* in these samples in relation to age (Fig. S2A). There is a positive correlation of Flower Lose and Age in both cohorts. We thank the reviewer for pointing us towards this important analysis. ($p = 3e-04$ for disease-free patients and $p = 1e-05$ for patients with one comorbidity).

Fig. S2A (fragment). *hFwe-Lose* expression increases in the lung tissue with age

i. Confusion regarding the authors model of how *Fwe* functions as a marker of fitness in cell-cell interactions. "...expression of the fitness marker *hFwe* was upregulated in lung tissue when compared with other examined tissues and that aged lung tissue had an upregulation of *hFweLose*, which correlates with a higher number of suboptimal cells." There are several problems with this and similar statements regarding the expression of *hFwe* and *hFweLose* in this study. As the senior authors have claimed in previous studies, one cannot interpret meaning from the expression or effect of *hFweLose* out of context - it is necessary to know the *Fwe* isoform status is in nearby cells. for example, from the Madan et al 2019, which is cited often in this work, "cells expressing *hFWE1* or *hFWE3* undergo cell death only when co-cultured with cells expressing either *hFWE2* or *hFWE4*". The authors have made a similar point in their publications using *Drosophila*: *hFweLose* only results in cell autonomous cell death when in presence of the other *Fwe* isoforms. Without quantifying the level of 'non-lose' isoforms of *hFwe*, it is impossible to know if *hFweLose* expression in the different tissues examined here is significant or of any consequence. In addition, the model presented by the authors on pages 6-7 does not make sense. According to their published model of *Flower* (Madan 2019, Rhiner 2010, etc), *hFweLose* doesn't need a SarsCoV2 infection to kill cells - in the right context, *hFweLose* is supposed to do that on its own.

Response

The comments of this Reviewer are welcomed, and we understand the reason why these comments are made. It is most likely that the first version of this manuscript has not properly explained our working hypothesis. In the revised version we have made more clear statements in the discussion to get rid of similar confusion. The comment of the Reviewer stems from the initial discovery, Madan et al, Nature 2019, where we established a *Flower-based* cell-cell recognition and cell fitness comparison system via expression of *hFwe-Win* and *Lose* isoforms. The elimination of suboptimal cells with low fitness which express *hFwe-Lose* by neighboring high fitness *hFwe-Win* expressing cells is an active process that occurs throughout the life cycle of an individual in various organs including lungs. With increasing age and other factors such as comorbidities, there seems to be a clear accumulation of low fitness cells (presented in our revised data). This phenomenon was also observed in *Drosophila* wing-discs where this cell-cell recognition and cell fitness comparison appeared to halt in clonal regions where a high number of low fitness cells were present (Levayer et al, Nature 2015). This essentially means that the elimination of low

fitness cells by neighboring high fitness cells requires the background tissue not have too many low fitness cells. In such an event this process will slow down otherwise most of the tissue will self-destruct.

We respectfully communicate to the Reviewer that this manuscript does not originate and depend on the concept of elimination of low fitness *hFwe-Lose* expressing cells at the hands of high fitness *hFwe-Win* expressing cells. We are not claiming that SARS-CoV2 infection is altering the interaction of cell competition-based interactions. In fact, we are claiming that SARS-CoV2 has nothing to do with cell competition, cell fitness comparisons, or expression of Flower Win and Lose isoforms. This manuscript is simply claiming that with increased age and acquired comorbidities or due to the underlying biology of any given individual, there appears to be an accumulation of cells with low fitness. And *hFwe-Lose* just tends to be a biomarker that is expressed in such cells with low fitness. We would like the Reviewer to carefully note that the elimination of low fitness cells is not because they have *hFwe-Lose* expression and that they are surrounded by cells expressing *hFwe-Win*. We are hypothesizing that suboptimal cells with low fitness tend to have poor robustness and are more likely to die under influence of external stimuli such as SARS-COV2 infection. It just so happens that such suboptimal low fitness cells tend to express *hFwe-Lose* as a biomarker and in this manuscript, we are using the expression of *hFwe-Lose* as a readout of the fitness status of the lung tissue in any individual.

j. Some arguments made by the authors are confusingly circular. For example, they state "Yet, although comorbidities are associated with increased case severity, patient risk is confounded by many variables. Therefore, the presence of co-existing disease and age alone cannot predict viral disease severity." The logic here seems to be flawed: they argue that *hFweLose* is a good fitness marker for Covid-19 because its expression is associated with COVID-19 comorbidities, while at the same time arguing that such comorbidities are not enough to predict disease severity.

Response

We respect the comments made by the Reviewer and we understand given the lack of solid data a question may arise that age and comorbidity are somewhat good indicators of COVID-19 severity outcomes, then why do we even need the Biomarker. The revised data with 283 retrospective and prospective patient cohorts answers this question. Expression of *hFwe-Lose* and tissue fitness are complex. It appears that they depend on age and comorbidities, but these do not appear to be the only factors that regulate this process. This is something which may be discovered in times to come. As the data stands today *hFwe-Lose* expression which signifies accumulation of suboptimal cells in respiratory tract tissue is a superior and high-efficiency biomarker (when compared with age and comorbidities) when it comes to predicting death as an outcome.

Minor concerns that should be addressed:

1. Were the disease-free controls also from autopsies? This is not made clear. Also not clear but relevant is the cause of the disease-free patient deaths and any compromising factors (e.g., non-smokers vs smokers, lung trauma, etc) they might have had.

Response

The controls are also from autopsy samples, and the included patients had no history of any other reported disease.

2. The legend to Fig. 1D says that tissues are "matched". The implication here is that each stromal area is matched to the tumor it surrounds it. How can this stroma be matched to normal lung (where there is no tumor)?

Response

As suggested by this reviewer and others this data is removed from the revised MS.

3. Fig. 2F legend does not match the figure.

Response

As suggested by this reviewer and others, this data is removed from the revised MS.

4. What were the contributions of each author?

Response

We have included a paragraph regarding the author contributions at the end of the revised manuscript.

5. Many references are incomplete (no journal, date, etc).

Response

We have corrected all references.

6. The methods are very sparsely written, making it difficult to understand how many of the analyses were actually done.

Response

We thank the reviewer for this comment. We have updated our methods part of the manuscript in order to make them more clear.

Referee #2:

This manuscript by Yekelchik et al. examines the value of hFweLose as a fitness mark in COVID -19. This study is from a group that has been studying Flower proteins in the context of cell fitness and apoptosis for some time; first in Drosophila and later in human cancer. The overall hypothesis that susceptible individuals with predisposing conditions have more "unfit" cells is reasonably interesting even if very descriptive.

This manuscript would be greatly improved by removing about 40% of the content, which is not original or meaningful and explaining things in more detail especially in the legends.

Response

We thank the reviewer and we agree with him. We have removed the unnecessary data in the revised version of the MS as pointed by this reviewer and the others.

1. Figure 1 A is quite obviously a schematic, but please state that in the legend.

Response

We have updated the legend for the Fig. 1A.

2. Why the emphasis on tumors in Figure 1? Why is this relevant in a manuscript about COVID-19? This topic was covered well in the Madan et al. paper in Nature in 2019 from the same group. No really relevant new information here, and the data here would be more appropriate for another cancer related paper. 3. Figure 1B is totally informative, in no way reflects "cross-talk" as labeled and should be deleted. 4. Figure 1E is a schematic which adds no information to concepts that could be easily gleaned from the data in Figure 1D. This issue has been covered before and does not belong in this manuscript. 5. The information in Figure 1F could be easily just covered by referring to the literature and does not belong in a primary paper.

Response

As suggested by this reviewer and others this data is removed from the revised MS. In the revised version of the manuscript, we excluded parts of Figure 1, which related to the analysis of publicly available datasets or referred to the commonly available

COVID-19 comorbidity statistics (namely, old parts B, C, D, E, F, G). Instead, we included references to the respective studies in the text.

3. The information in the figure legends needs to be made more clear and with explicit information as to which data bases information was obtained for each panel from 1G to 1N.

Response

As suggested by this reviewer and others we have removed the plots, related to the data from databases and/or already published data (previously Fig. 1B,C,D,E,F,G,J,K,L), and only left plots, based on our novel experimental data. We now also included clearer legends for each plot.

4. The statement about the "largest" autopsy series with just 11 patients is odd. The legend to figure 2A suggests that 249 COVID patients were studied. What is the real information here? There needs to be far more information right through-everything is very cryptic and poorly explained.

Response

At the time the previous version of the manuscript was written Menter et al., 2020, was the study which included one of the largest robotic autopsies for COVID-19 patients. As suggested by the reviewer we have removed these lines. We have made the legends more clear for the modified figures.

8. Including the CT scans from 2 patients is totally unnecessary. Plenty of published CT scans already. Fig 2F should be deleted.

Response

We have deleted the CT scans from Figure 2.

9. Showing the apoptosis in 2G could be considered relevant - please show high power views.

Response

As suggested by the reviewer, higher magnification images showing apoptosis have been added.

Referee #3:

The study by Michail Yekelchik, Esha Madan and colleagues investigates a possible correlation between cell competition failure and Covid-19 severity. In particular, they analyse the expression of a well-characterised marker of cell competition, Flower (Fwe), whose Lose isoform (hFweLose) is known to tag suboptimal cells for apoptotic death in human cancers. Apoptosis is a major challenge in MERS and SARS, and the consistent tissue damage results in severe adverse symptoms. The main finding of this study is that hFweLose expression increases with ageing, hypertension, diabetes, obesity and other conditions considered comorbidity factors in Covid-19 patients. The presence of one or more of these conditions is known to exacerbate the disease, and this correlates with a consistent increase of hFweLose expression, setting it as a bona-fide predictive marker for disease outcome.

The study is of high interest as it unveils a novel, relevant role for cell competition (more generally, for homeostatic tissue regulation) in the response to infectious disease. The authors hypothesise that healthy lungs (such as other organs) be capable to promptly identify and eliminate unfit cells, so maintaining organ fitness at the best, but the presence of too many suboptimal cells in ageing or sick lungs disrupts this process, and allows accumulation of unfit cells, making lungs susceptible to a worse disease progression. Although the reason why elders, along with fragile patients, respond poorly to SARS-CoV-2 aggression seems obvious at the intuitive level, findings are missing about specific mechanisms and molecules which can underlie

individual vulnerability. The authors are a distinguished voice in the topic of cell competition and their premises are convincing. Methods are briefly but adequately described, results are clear, statistical analysis is appropriate and the overall conceptual and experimental scheme is robust.

The manuscript necessitates some minor amendments before being accepted for publication, as follows:

a. Results, page 3:

Please insert: (Fig. 1E) after the sentence: This elimination of stromal cells creates space and availability of nutrients for cancer to grow and dominate the tissue space.

Response

We thank the reviewer for this remark. In the revised version of the manuscript, we excluded panel 1E and now instead provide a reference to a published study regarding the mechanism of Flower-mediated cell competition.

b. Results, page 5:

In the second line the authors refer to 21 patients and cite the same study as earlier in the text, where they refer to 11 patients. Please conform patient number between the two statements.

Response

Inconsistencies in patient numbers have been corrected.

c. Fig. 1F ICU=Intensive Care Unit, please specify it in the legend.

Response

As suggested by the reviewer the abbreviation ICU is appropriately explained in both the manuscript text and the legends.

d. Fig. 2F Numbers and letters in the figure legend do not match with those reported in the figure. Please fix it.

Response

We have corrected all references to Figure numbers within the text body and legends.

e. Fig. 2G Readers may not be confident with lung histology, so please include a detailed frame of a normal lung specimen. In G1, i = Cell elimination: please explain how you recognise the encircled cells are being eliminated. In G2, does "The adjacent interstitial space shows analogous changes to 2B1" refer to Figure "2F"? See above.

Response

We thank the reviewer for this comment and we have included a detailed frame of a normal lung specimen. We have explained how we recognise the elimination of the encircled cells and we fixed the legend of previous figure G2.

f. Fig. 2H: please indicate with an arrow or similar the two apoptotic cells in the upper left alveolus. The legend for FweLose expression analysis in Cas-pos and Cas-neg cells is completely missing.

Response

We have included arrows to indicate apoptotic cells in previous Fig. 2H and have included the legends for the other figure.

Summary:

In the concluding sentence, the authors claim FweLose expression may be readily detected in patient samples, but how do they think to assess it in Covid-19 patients? What tissues/cells should be processed to get reliable risk predictions? The authors should discuss this, or mitigate this affirmation. Some typos are found across the text, please go carefully through it and fix them.

Response

We have now assessed *hFwe-Lose* RNA expression within nasal epithelial cells harvested from patient nasal swabs collected for standard COVID-19 testing. The text has been thoroughly edited during revision.

Referee #4:

The authors have done an extremely rigorous evaluation of flower lose expression in lung tissue as a predictor of Covid -19 severity and prognosis. Unfortunately, they have failed to perform the one clinical correlation that has prognostic meaning e.g. measuring flower lose expression in peripheral blood and correlating that with expression in the lung and clinical outcome in an actual prospective study. Autopsy studies and cellular studies are of some value but what every clinician wants to know is whether the Flower Lose expression adds to clinical predictors of outcome and hence can be used to justify early and aggressive treatment. Based on this standard, calling this a clinical marker of prognosis is overreaching.

Response

We thank the reviewer for the comments and for pin-pointing the important shortcomings of our original manuscript. We worked hard to overcome them, so we included a lot of new data and completely overhauled the manuscript. Particularly, we removed the parts, related to analysis of already published datasets, and focused on our original research. Most importantly, we now included a large cohort of nasal swab samples from COVID-19 patients, and thoroughly investigated the potential of *hFwe-Lose* expression to predict these patients' disease outcome. Furthermore, we separated all patients into training and testing cohorts, and compared the predictive potential of *hFwe-Lose* to commonly used clinical biomarkers, as well as patients' age and comorbidities. We markedly reinforced our biostatistical analysis as well, and now we provide multiple complementary state-of-the-art clinical statistical tests, which all highlight the superiority of *hFwe-Lose* in predicting worse COVID-19 disease outcome.

Major comment #1 what you have done is shown that Flower Lose expression correlates with clinical events in vitro but you have no evidence of its predictive value in vivo in living subjects. I think you need to tone down the predictive claims here since there is absolutely no evidence of prediction in a clinical scenario in a living subject.

Response

In the revised manuscript, we have analyzed the expression of *hFwe-Lose* in nasal swabs of living subjects and correlated its expression with patient clinical data. Our analysis shows that knowing *hFwe-Lose* expression in a nasal swab, as well as patients' age is sufficient to surpass the prediction accuracy of hospitalization and/or death, compared to commonly used clinical biomarkers.

Major comment #2: What is the relationship between Flower Lose expression in lung tissue and Flower Lose expression in the serum or plasma of a living patient with Covid-19 infection?

Response

We thank the reviewer for this suggestion. However, we decided to investigate the *hFwe-Lose* expression in even more readily available biological samples – in nasal swabs. The advantages of using swabs instead of blood samples are that they more accurately displaying the fitness of patients' lungs, compared to blood, and that they are very easy to collect. Furthermore, *hFwe-Lose* expression could be measured during the standard RT-qPCR COVID-19 test.

Major comment #3: Does Flower Lose expression in peripheral blood in a large cohort of Covid-19 infected patients have any predictive power with regard to clinical outcomes?

Response

In our nasal swabs' dataset, comprised of 203 patients from training and 80 patients of testing cohorts, we were able to reach a high predictive power of *hFwe-Lose* in regard to patients' outcome. Namely, we reached positive predictive values (PPV) of 83.7%/87.8% and negative predictive values (NPV) of 67.2%/64.1% (AUCROC = 0.89) for predicting hospitalisation (training/testing cohorts); and PPVs of 34.5%/100% and NPVs of 100%/93.2% (AUCROC = 0.98) for predicting death (training/testing cohorts) (Figures 3-4).

Major Comment #4: If the answer to comment #3 is yes is it independent of comorbidities e.g. age, BMI, sex, etc. These are the questions you need to answer. What you have shown is that there is an in vitro correlation of Flower Lose expression in the lung in patients who have died from Covid 19.

Response

We thank the reviewer for these important suggestions. We now specifically analysed *hFwe-Lose* expression in nasal swabs in order to determine the correlation between tissue fitness and COVID-19 disease severity. Data added upon revision includes expression of *hFwe-Lose* from a large cohort of patient nasal swabs, which is a standard collection method as part of COVID-19 testing. Thus, we have shown that *hFwe-Lose* can be readily detected using a practical and widely employed diagnostic assay. As suggested by the reviewer, we also included all patients' parameters (Table 1) into our biostatistical analysis (Figures 2-4).

11th Aug 2021

Dear Dr. Gogna,

Thank you for the submission of your revised manuscript to EMBO Molecular Medicine. I am pleased to inform you that we will be able to accept your manuscript pending the following final amendments:

1) In the main manuscript file, please do the following:

- Correct/answer the track changes suggested by our data editors by working from the attached document.
- Add up to 5 keywords.
- Make sure that all special characters display well.
- Add contributions for Ronny Drapkin and please use author initials.
- In M&M, statistical paragraph should reflect all information that you have filled in the Authors Checklist, especially regarding randomization, blinding, replication etc.
- Provide data availability statement. If no data are deposited in public repositories, please add the sentence: "This study includes no data deposited in external repositories".

Please check "Author Guidelines" for more information.

<https://www.embopress.org/page/journal/17574684/authorguide#availabilityofpublishedmaterial>

2) Appendix: Please correct nomenclature to "Appendix Figure S1" etc. and "Appendix Table S1" etc.

3) Funding: Please make sure that information about all sources of funding (including grant numbers) are complete in both our submission system and in the manuscript.

4) Source data: We encourage you to include the source data for figure panels that show essential data. Numerical data should be provided as individual .xls or .csv files (including a tab describing the data). For blots or microscopy, uncropped images should be submitted (using a zip archive if multiple images need to be supplied for one panel). Please check "Author Guidelines" for more information. <https://www.embopress.org/page/journal/17574684/authorguide#sourcedata>

5) The Paper Explained: Please provide "The Paper Explained" and add it to the main manuscript text. Please check "Author Guidelines" for more information.

<https://www.embopress.org/page/journal/17574684/authorguide#researcharticleguide>

6) Synopsis: Every published paper now includes a 'Synopsis' to further enhance discoverability. Synopses are displayed on the journal webpage and are freely accessible to all readers. They include separate synopsis image and synopsis text.

- Synopsis image: Please provide a striking image or visual abstract as a high-resolution jpeg file 550 px-wide x (250-400)-px high to illustrate your article.

- Synopsis text: Please provide a short standfirst (maximum of 300 characters, including space) as well as 2-5 one sentence bullet points that summarise the paper as a .doc file. Please write the bullet points to summarise the key NEW findings. They should be designed to be complementary to the abstract - i.e. not repeat the same text. We encourage inclusion of key acronyms and quantitative information (maximum of 30 words / bullet point). Please use the passive voice.

7) For more information: There is space at the end of each article to list relevant web links for further consultation by our readers. Could you identify some relevant ones and provide such information as well? Some examples are patient associations, relevant databases, OMIM/proteins/genes links, author's websites, etc...

8) Press release: Please inform us as soon as possible and latest at the time of submission of the

revised manuscript if you plan a press release for your article so that our publisher could coordinate publication accordingly.

9) Please be aware that we use a unique publishing workflow for COVID-19 papers: a non-typeset PDF of the accepted manuscript is published as "Just Accepted" on our website. With respect to a possible press release, we have the option to not post the "Just Accepted" version if you prefer to wait with the press release for the typeset version. Please let us know whether you agree to publication of a "Just accepted" version or you prefer to wait for the typeset version.

10) As part of the EMBO Publications transparent editorial process initiative (see our Editorial at <http://embomolmed.embopress.org/content/2/9/329>), EMBO Molecular Medicine will publish online a Review Process File (RPF) to accompany accepted manuscripts. This file will be published in conjunction with your paper and will include the anonymous referee reports, your point-by-point response and all pertinent correspondence relating to the manuscript. Let us know whether you agree with the publication of the RPF and as here, if you want to remove or not any figures from it prior to publication. Please note that the Authors checklist will be published at the end of the RPF.

11) Please provide a point-by-point letter INCLUDING my comments as well as the reviewer's reports and your detailed responses (as Word file).

I look forward to reading a new revised version of your manuscript as soon as possible.

Yours sincerely,

Zeljko Durdevic

***** Reviewer's comments *****

Referee #2 (Comments on Novelty/Model System for Author):

Even if the relevance of this work was already clear to me while reading the previous version, the revised manuscript has addressed a series of major and minor concerns, above all the lack of information about how to make Fwe an effective prediction marker for Covid-19 severity. The prospective study included in the revised version indicates Fwe expression can be assessed in samples from routine nasal swabs. Besides this, the present version has been tremendously improved in patient recruitment, data correlation and statistical analysis and, in my opinion, it is worth immediate publication.

Referee #3 (Comments on Novelty/Model System for Author):

No comments at this time

Referee #3 (Remarks for Author):

Suitable for publication

Dear Editors
EMBO Molecular Medicine,

We are submitting the revised version of our manuscript titled "Flower Lose, a Cell Fitness Marker, Predicts COVID-19 Prognosis", for your kind consideration for publication.

Firstly, we would like to thank you and the Reviewers for each comment and efforts to improve the quality of this manuscript.

We corrected and edited the manuscript taking into consideration every suggestion and concern made by you and the Reviewers. Below, we provide a detailed point-by-point response to yours and reviewers comments:

***** Editor's comments *****

1) In the main manuscript file, please do the following:

- a. Correct/answer the track changes suggested by our data editors by working from the attached document.
R – We revised the whole manuscript and added all the information requested by the reviewers. All added information is marked in red to highlight our answers to reviewer's suggestions.
- b. Add up to 5 keywords.
R – We have added the keywords.
- c. Make sure that all special characters display well.
R – All special characters are well displayed.
- d. Add contributions for Ronny Drapkin and please use author initials.
R – We added contribution for Ronny Drapkin using author initials.
- e. In M&M, statistical paragraph should reflect all information that you have filled in the Authors Checklist, especially regarding randomization, blinding, replication etc.
R – We added a statistical paragraph in M&M which reflects all information that we have filled in the Authors Checklist regarding randomization, blinding, replication, etc.
- f. Provide data availability statement. If no data are deposited in public repositories, please add the sentence: "This study includes no data deposited in external repositories".
R – We have provided the statement in the MS.

2) Appendix: Please correct nomenclature to "Appendix Figure S1" etc. and "Appendix Table S1" etc

R – We corrected the nomenclature of supplementary figures and tables in the submission system.

3) Funding: Please make sure that information about all sources of funding (including grant numbers) are complete in both our submission system and in the manuscript.

R – We revised all source of funding, assuring that their information is complete and the same in the submission system and the manuscript.

4) Source data: We encourage you to include the source data for figure panels that show essential data. Numerical data should be provided as individual .xls or .csv files (including a tab describing the data). For blots or microscopy, uncropped images should be submitted (using a zip archive if multiple images need to be supplied for one panel). Please check "Author Guidelines" for more information.

<https://www.embopress.org/page/journal/17574684/authorguide#sourcedata>

R – We have added all information in the tables associated with the MS.

5) The Paper Explained: Please provide "The Paper Explained" and add it to the main manuscript text. Please check "Author Guidelines" for more information.

<https://www.embopress.org/page/journal/17574684/authorguide#researcharticleguide>

R – We added The Paper Explained section in the main manuscript text following “Author Guidelines”.

- 6) Synopsis: Every published paper now includes a 'Synopsis' to further enhance discoverability. Synopses are displayed on the journal webpage and are freely accessible to all readers. They include separate synopsis image and synopsis text.
- Synopsis image: Please provide a striking image or visual abstract as a high-resolution jpeg file 550 px-wide x (250-400)-px high to illustrate your article.
 - Synopsis text: Please provide a short standfirst (maximum of 300 characters, including space) as well as 2-5 one sentence bullet points that summarise the paper as a .doc file. Please write the bullet points to summarise the key NEW findings. They should be designed to be complementary to the abstract - i.e. not repeat the same text. We encourage inclusion of key acronyms and quantitative information (maximum of 30 words / bullet point). Please use the passive voice.

R – We added in the submission system the synopsis image and text following the mentioned guidelines.

- 7) For more information: There is space at the end of each article to list relevant web links for further consultation by our readers. Could you identify some relevant ones and provide such information as well? Some examples are patient associations, relevant databases, OMIM/proteins/genes links, author's websites, etc...

R – We do not have more relevant information for further consultation.

- 8) Press release: Please inform us as soon as possible and latest at the time of submission of the revised manuscript if you plan a press release for your article so that our publisher could coordinate publication accordingly.

R – We would like to inform you that we plan to have a press release of our article.

- 9) Please be aware that we use a unique publishing workflow for COVID-19 papers: a non-typeset PDF of the accepted manuscript is published as "Just Accepted" on our website. With respect to a possible press release, we have the option to not post the "Just Accepted" version if you prefer to wait with the press release for the typeset version. Please let us know whether you agree to publication of a "Just accepted" version or you prefer to wait for the typeset version.

R – We kindly ask you to not post the "Just Accepted" version of the paper.

- 10) As part of the EMBO Publications transparent editorial process initiative (see our Editorial at <http://embomolmed.embopress.org/content/2/9/329>), EMBO Molecular Medicine will publish online a Review Process File (RPF) to accompany accepted manuscripts. This file will be published in conjunction with your paper and will include the anonymous referee reports, your point-by-point response and all pertinent correspondence relating to the manuscript. Let us know whether you agree with the publication of the RPF and as here, if you want to remove or not any figures from it prior to publication. Please note that the Authors checklist will be published at the end of the RPF.

R – If it is possible and allowed we would not like to publish the Review Process File. -> Author agreed 16.09.2021

***** Reviewer's comments *****

Referee #2 (Comments on Novelty/Model System for Author):

Even if the relevance of this work was already clear to me while reading the previous version, the revised manuscript has addressed a series of major and minor concerns, above all the lack of information about how

to make Fwe an effective prediction marker for Covid-19 severity. The prospective study included in the revised version indicates Fwe expression can be assessed in samples from routine nasal swabs. Besides this, the present version has been tremendously improved in patient recruitment, data correlation and statistical analysis and, in my opinion, it is worth immediate publication.

Referee #3 (Comments on Novelty/Model System for Author):

No comments at this time

Referee #3 (Remarks for Author):

Suitable for publication

R – We took in highest consideration all the reviewers comments and suggestions, and we implemented changes in our study accordingly. We hope that those changes met all the reviewers concerns and that you find this manuscript relevant for publication. We again thank all the contribution that those comments and suggestions had to significantly improve the quality of our manuscript and we hope this study can improve the scientific knowledge regarding COVID-19 pandemic.

16th Sep 2021

Dear Dr. Gogna,

We are pleased to inform you that your manuscript is accepted for publication and is now being sent to our publisher to be included in the next available issue of EMBO Molecular Medicine.

Corresponding Author Name: Dr. Rajan Gogna

Manuscript Number: EMM-2020-13714-V2